# On Fairness of Task Arithmetic: The Role of Task Vectors

**Laura Gomezjurado Gonzalez**[1][♯][*]   **Hiroki Naganuma**[2,3][♯][†]   **Kotaro Yoshida**[4][♯][‡]
**Yuji Naraki**[5]   **Takafumi Horie**[6]   **Ryotaro Shimizu**[7]
[1]Stanford University   [2]Mila   [3]Université de Montréal   [4]Institute of Science Tokyo
[5]Independent Researcher   [6]Kyoto University   [7]ZOZO Research

## Abstract

Model editing techniques, particularly task arithmetic with task vectors, offer an efficient alternative to full fine-tuning by enabling direct parameter updates through simple arithmetic operations. While this approach promises substantial computational savings, its impact on fairness has remained largely unexplored—despite growing concern over biased outcomes in high-stakes applications such as hate speech detection. In this work, we present the first systematic study of *group fairness* in task arithmetic within this binary text and image classification regime, comparing it against full fine-tuning (FFT) and Low-Rank Adaptation (LoRA). We evaluate across multiple language models and datasets using standard group fairness metrics, including Demographic Parity and Equalized Odds. Our analysis shows that task vectors can be tuned to achieve competitive accuracy while reducing disparities, and that merging subgroup-specific task vectors provides a practical mechanism for steering fairness outcomes. We further provide a theoretical bound linking task vector scaling to fairness metrics, offering insight into the observed trade-offs. Together, these findings establish task arithmetic not only as a cost-efficient editing method but also as a fairness-aware alternative to existing adaptation techniques, within the standard group-fair classification setting, laying the groundwork for responsible deployment of large language models. Our code is available at `https://github.com/LauraGomezjurado/fairness_task_vector_deploy`

## 1 Introduction

As large language models (LLMs) are deployed across increasingly diverse applications, efficient techniques for adapting them to specific tasks have become essential. While model distillation and compact architectures reduce computational demands (Sanh et al., 2019b; Jiao et al., 2020; Turc et al., 2020; Abdin et al., 2024), task-specific fine-tuning (FFT) remains resource-intensive. This has motivated parameter-efficient fine-tuning (PEFT) methods such as adapters and Low-Rank Adaptation (LoRA) (Houlsby et al., 2019; Hu et al., 2022; Ben Zaken et al., 2022; Dettmers et al., 2023), which update only a small fraction of parameters.

LoRA exemplifies this trade-off: it preserves most of the pretrained weights while reducing training costs. However, PEFT methods do not resolve deeper concerns. In high-stakes domains with imbalanced data—such as toxicity or hate-speech detection—they can maintain or even amplify biases (Ding et al., 2024; Sap et al., 2019), raising concerns about fairness.

A promising alternative is task arithmetic with task vectors (Ilharco et al., 2023; Zhang et al., 2024; Yoshida et al., 2025b; Yoshikawa et al., 2025; Yoshida et al., 2025a). A task vector is defined as the difference between a fine-tuned model and its base counterpart. By adding, subtracting, or scaling such vectors, one can directly edit model behavior without gradient-based retraining. This approach

---

[*]Corresponding author lpgomez@stanford.edu
[†]Corresponding author naganuma.hiroki@mila.quebec.
[‡]Corresponding author yoshida.k.0253@m.isct.ac.jp
[♯]These authors contributed equally to this work.

offers (i) computational efficiency, (ii) fine-grained control over transferred capabilities, and (iii) enhanced interpretability when task vectors are associated with specific subgroups (Cerrato et al., 2022). Yet its fairness implications remain poorly understood. For example, enhancing performance on one demographic subgroup may inadvertently degrade outcomes for another, and the trade-offs with standard metrics such as Demographic Parity (DPD) or Equalized Odds (EOD) remain unclear.

To address this gap, we conduct the first systematic study of fairness in task arithmetic. We compare task vector editing against both FFT and LoRA, and we further investigate whether injecting subgroup-specific task vectors into an FFT model provides additional control over fairness outcomes. In the NLP domain, our experiments focus on hate-speech detection with LLaMA-7B (Touvron et al., 2023) and toxicity detection on Civil Comments (Borkan et al., 2019) using DistilBERT and Qwen2.5-0.5B (Qwen Team, 2025). In the computer-vision domain, we evaluate age classification on UTKFace (Zhang et al., 2017) using ViT-Base/16 (Dosovitskiy et al., 2021). Across both domains and model architectures, we observe consistent fairness–utility trade-offs.

Our contributions are as follows:

- **Comprehensive evaluation**: We compare FFT, LoRA, task vector editing, and a hybrid approach that injects task vectors into FFT, analyzing their impact on fairness metrics and predictive performance (Figure 1).

- **Fairness through scaling**: We show that adjusting task vector coefficients can substantially improve fairness while maintaining accuracy (Figure 2).

- **Subgroup-sensitive editing**: We demonstrate that merging task vectors from underrepresented subgroups allows targeted fairness adjustments with negligible accuracy loss (Figures 3a, 3b, 4a).

- **Theoretical grounding**: We derive an upper bound linking task vector scaling to DPD and EOD, providing a principled explanation for the observed fairness–accuracy trade-offs (Section 5.1 and Appendix C).

Throughout, the prediction task remains binary, but the fairness structure is not: we evaluate disparities across multiple demographic groups, following widely used group-fairness benchmarks (Hardt et al., 2016; Agarwal et al., 2018; Zafar et al., 2017; Kearns et al., 2018; Das et al., 2024; Dutt et al., 2023; Sukumaran et al., 2024). Taken together, our results provide an empirical foundation for understanding fairness–accuracy trade-offs in task arithmetic within the standard group-fair binary-classification regime, while pointing toward extensions to more complex tasks (e.g., multi-label or generative settings) as future work.

## 2 PRELIMINARIES

In this section, we first provide an overview of the fundamental concept of task vectors and the procedure known as task arithmetic, which applies these vectors to edit model behavior. We then introduce methods for merging multiple task vectors into a single model.

**Task arithmetic.** A task vector is defined as the difference in model parameters between a fine-tuned model on a given task and the original base model. Formally, if $\theta_{\text{base}}$ are the pre-trained weights and $\theta_{\text{task}}$ are the weights after fine-tuning on a task, then the task vector is: $\Delta\theta = \theta_{\text{task}} - \theta_{\text{base}}$ (Ilharco et al., 2023).

This vector represents a direction in weight space such that moving the base model's weights by $\Delta\theta$ steers the model to perform well on that task. In other words, adding $\Delta\theta$ to $\theta_{\text{base}}$ yields a model with improved performance on the target task, without any additional training. Once computed, task vectors can be manipulated through simple arithmetic operations to edit model behavior directly in weight space (Ilharco et al., 2023; Ortiz-Jimenez et al., 2024). Key operations include:

> **Addition:** Given two task vectors $\Delta\theta_A$ and $\Delta\theta_B$ (for tasks A and B), their sum can be applied to the base model ($\theta_{\text{base}} + \Delta\theta_A + \Delta\theta_B$) to produce a model that exhibits improved performance on both tasks A and B (Ilharco et al., 2023). This task addition effectively combines knowledge from multiple tasks into one model.

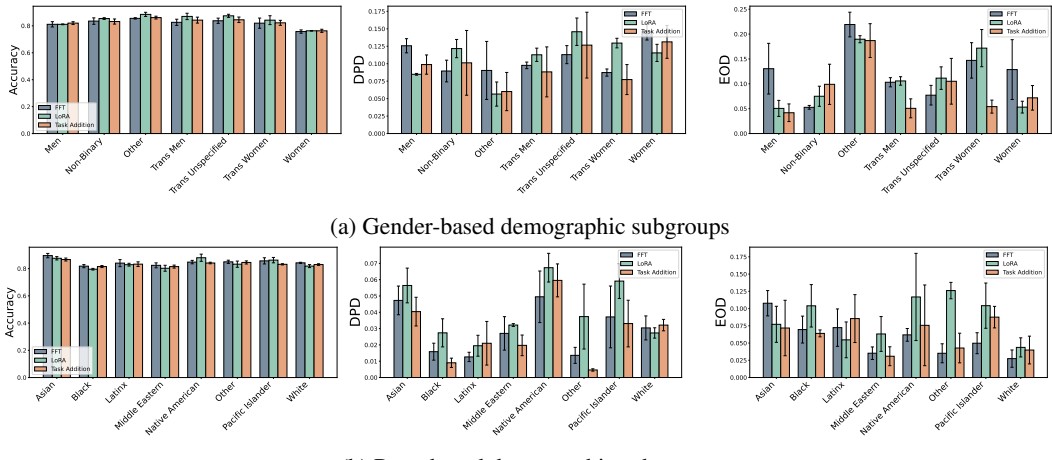

(a) Gender-based demographic subgroups

(b) Race-based demographic subgroups

Figure 1: LoRA and FFT vs. Task addition with the optimal coefficient for the training accuracy ($\lambda = 0.8$ for gender setting and $\lambda = 0.5$ for race setting) on group-wise accuracy, demographic parity difference (DPD, lower is fairer), and equalized odds difference (EOD, lower is fairer). Error bars denote the standard error across three seeds. Columns: group-wise accuracy, DPD, EOD. No consistent pattern emerges indicating that task addition systematically degrades subgroup fairness relative to LoRA or FFT. While some subgroups show improvements or comparable results under task addition, others exhibit small declines.

> **Negation:** Using the negative of a task vector, $-\Delta\theta$, one can subtract a task's influence. For example, applying $\theta_{\text{base}} - \Delta\theta_A$ (or equivalently $\theta_{\text{base}} + (-\Delta\theta_A)$) yields a model with reduced performance on task A—effectively unlearning or forgetting it—while preserving other behaviors (Ilharco et al., 2023). This is useful for removing undesirable skills or biases.
>
> **Scalar scaling:** Multiplying a task vector by a scalar $\lambda$ adjusts the strength of the edit. For example, using $\theta_{\text{base}} + \lambda\Delta\theta_A$ allows partial ($0 < \lambda < 1$) or amplified ($\lambda > 1$) application of a task's effect. This scaling provides fine-grained control over how strongly the task knowledge is injected into the model.

**Merging task vectors.** Because task vectors live in a common weight space, they can be merged by linear combination. Let $\theta_0$ denote a base model and let $\{\Delta\theta_i\}_{i=1}^K$ be task vectors. A generic merged model takes the form

$$\theta_{\text{merged}} = \theta_0 + \sum_{i=1}^K \lambda_i \Delta\theta_i, \tag{1}$$

where the coefficients $\lambda_i$ govern the relative contribution of each task (Zhang et al., 2024). Prior work explores learning these coefficients with, for example, Bayesian optimization or multi-objective search over $\boldsymbol{\lambda} = (\lambda_1, \ldots, \lambda_K)$, but such methods introduce additional computational cost and stochasticity.

In this work we deliberately adopt the simpler and widely used *single-scalar* parameterization, in which a global coefficient $\lambda$ scales the entire task vector update. Concretely, the models we evaluate can be written as

$$\theta(\lambda) = \theta_0 + \lambda \, \Delta\theta, \tag{2}$$

where $\Delta\theta$ is a fixed task vector (or sum of task vectors), and the same scalar $\lambda$ is applied uniformly across all inputs, demographic subgroups, and fairness metrics. This single global coefficient mirrors how task-arithmetic toolkits are exposed in practice, and it provides a one-dimensional, easily interpretable control knob for tracing fairness–accuracy trade-offs. We treat $\lambda$ as a hyperparameter and select it by grid search over a fixed set of candidate values on a held-out validation set, using a common objective that balances accuracy and group-fairness measures. The same validation protocol is used to tune the corresponding regularization weights for our SFT, LoRA, and task-arithmetic models, ensuring that all methods are compared under matched fairness–accuracy trade-offs.

## 3 RELATED WORK

This section outlines how FFT, LoRA, task merging, and fairness are related. Further discussion of task-arithmetic efficiency, interpretability, and fairness in VLMs appears in Appendix A.

**FFT and LoRA under fairness constraints.** Parameter-efficient methods such as LoRA (Hu et al., 2022) address computational bottlenecks by training only a small subset of parameters, yet they do not inherently resolve fairness concerns. Empirical studies show that LoRA's impact on subgroup performance is mixed: in some settings it achieves results comparable to full fine-tuning (Ding et al., 2024), while in others it preserves undesirable biases or fails to mitigate toxic behaviors (Das et al., 2024). These divergent outcomes depend on factors such as the rank of the LoRA updates, the base model's pre-existing biases, and the distributional properties of the fine-tuning data (Das et al., 2024). Recent work begins to incorporate fairness objectives directly into PEFT. Wang and Demberg (Wang & Demberg, 2024) introduce a multi-objective method that jointly targets accuracy and reduced stereotypical bias, and Sukumaran et al. (Sukumaran et al., 2024) propose FairLoRA, which embeds fairness-driven structure into low-rank updates for vision models. These efforts show that fairness-aware PEFT is feasible but often requires custom objectives and careful tuning.

**Merging tasks and fairness composition.** Despite the potential efficiency gains and interpretability offered by task arithmetic, the merging of task vectors for multiple groups can trigger new challenges. For instance, simply summing vectors may lead to "negative transfer," where updates beneficial to one subgroup degrade performance for another (Ding et al., 2023; Yu et al., 2020). In highly imbalanced settings, merging models through supervised fine-tuning can also disproportionately favor majority groups while disadvantaging minorities (Cross et al., 2024).

Additionally, prior work shows that fairness guarantees often do not compose: even if individual components satisfy group or individual fairness in isolation, composing them can break those guarantees (Dwork & Ilvento, 2018). This motivates our focus on post-hoc task-arithmetic edits: adding or scaling subgroup task vectors can be viewed as composing behaviors, and interactions among subgroup-specific task vectors can produce unpredictable shifts in metrics like Demographic Parity and Equalized Odds (Gohar et al., 2023). Consequently, identifying effective ways to adjust task vectors—such as through scalar scaling—remains a key step toward fairness-aware model editing. This work aims to fill that gap by systematically evaluating how these operations influence both fairness and overall model accuracy.

In parallel, multi-task fairness methods such as Multi-Task-Aware Fairness (Wang et al., 2021), Learning-to-Teach Fairness-Aware MTL (Roy & Ntoutsi, 2022), and FairBranch (Roy et al., 2024) manage fairness–accuracy trade-offs during training. Our study complements these by asking: when we edit models *after training* via task vectors, can simple controls (e.g., $\lambda$-scaling) recover fairer behavior without retraining?

**Group fairness metrics in binary text classification.** Research on group fairness in text classification predominantly adopts the binary prediction setting (e.g., toxic vs. non-toxic) with multiple demographic groups (Hardt et al., 2016; Agarwal et al., 2018; Zafar et al., 2017; Kearns et al., 2018; Das et al., 2024; Dutt et al., 2023; Sukumaran et al., 2024). Within this framework, two metrics have emerged as canonical due to their interpretability and broad adoption: Demographic Parity Difference (DPD), which measures disparities in positive prediction rates, and Equalized Odds Difference (EOD), which measures disparities in true- and false-positive rates (Hardt et al., 2016; Feldman et al., 2015; Kennedy et al., 2020a). These metrics form the standard baseline in fairness auditing toolkits (Bellamy et al., 2018; Fairlearn contributors, 2025) and underpin much of the fair-PEFT literature (Fraenkel, 2020; Pitoura, 2019; Quan et al., 2023; Ding et al., 2024). Their centrality is motivated by theory: for calibrated, score-based classifiers with group-agnostic thresholds, many group-fairness desiderata collapse to constraints on selection-rate and error-rate disparities, and classical impossibility results show that calibration, selection parity, and error parity cannot be simultaneously satisfied when base rates differ (Kleinberg et al., 2017). Consequently, DPD and EOD reveal the essential structure of fairness–accuracy trade-offs without invoking additional structural or counterfactual assumptions.

Our evaluation follows this binary prediction framework but focuses on a multi-group setting. We report both macro-averaged and worst-group versions of DPD and EOD. This choice allows us to

capture disparities across the full distribution of subpopulations, where fairness–accuracy tensions are often most pronounced. Fairness metrics for multiclass, multi-label, or generative models remain active research areas, and no widely accepted standard analogous to DPD/EOD currently exists. Given this landscape, the binary multi-group regime offers a well-defined and empirically robust foundation for systematic fairness analysis. Formal metric definitions appear in Appendix B, and implementation details are provided in §4.1.

## 4 EXPERIMENTAL SETUP

### 4.1 CONFIGURATION

Building on the experimental framework established by Ding et al. (2024), we adopted their evaluation and experimental procedure to assess the fairness implications of LoRA in comparison to FFT. In our work, we extend this analysis by focusing on how task arithmetic compares to both LoRA and FFT in terms of fairness and performance. The detailed experimental setup is provided in Appendix D.

| Gender Subgroups | | Race Subgroups | |
|---|---|---|---|
| Men | 817 | Asian | 311 |
| Non-binary | 114 | Black | 1,007 |
| Trans men | 178 | Latinx | 368 |
| Trans unspecified | 173 | Native American | 153 |
| Trans women | 148 | Middle Eastern | 493 |
| Women | 2,057 | Pacific Islander | 138 |
| Other | 59 | White | 580 |
| | | Other | 302 |
| **Total** | **3,546** | **Total** | **3,352** |

Table 1: Berkeley D-Lab Hate Speech data statistics in the gender and race subgroups.

**Datasets.** We build on the *Berkeley D-Lab Hate Speech* dataset introduced by Kennedy et al. (2020a) and adapted by Ding et al. (2024). Our version contains 6,898 tweet-length snippets annotated for hate speech and two sensitive attributes, *Race* and *Gender*, each with fine-grained subgroups (e.g., *Women*, *Non-binary*, *Men* for *Gender*; see Table 1). We treat hate speech detection as binary classification and use the subgroup annotations (e.g., *gender* = woman, *race* = Asian) to compute subgroup-level performance and fairness metrics. To test generalization beyond hate speech, we also evaluate on the *Civil Comments* dataset (Borkan et al., 2019), a large-scale toxicity corpus with sensitive-attribute labels. We binarize toxicity using a threshold of $0.5$, treating comments above this value as positive ("flagged"), and assess fairness across *Gender* and *Race* subgroups. For the vision domain, we follow Ding et al. (2024) and use the *UTKFace* face image dataset (Zhang et al., 2017), which provides *age* (in years), *gender*, and coarse-grained *race* labels. We convert it into a binary age-classification task by thresholding age at 30 years into "younger" ($\leq$30) vs. "older" ($>$30), with the older group as the positive class. We treat *race* and *gender* as sensitive attributes and evaluate subgroup performance across the standard UTKFace race categories (*White*, *Black*, *Asian*, *Indian*, *Others*) and gender categories (*Male*, *Female*), mirroring the subgroup-based analysis in Ding et al. (2024).

**Evaluation metrics and fairness scope.** Our fairness analysis follows the standard group-fairness framework for supervised classification, where two criteria form the canonical basis for auditing disparate impact across demographic groups. *DPD* captures disparities in selection rates, and *EOD* captures disparities in class-conditional error rates (TPR/FPR). These metrics are widely used, theoretically grounded, and highly interpretable: together they summarize the allocation and error-rate harms that motivate much of the fairness literature. This makes the binary text and image classification setting—where selection and error rates are well-defined—the natural domain in which to study fairness effects of task vector editing. For each protected attribute (e.g., gender, race/ethnicity), we compute subgroup-resolved DPD, EOD, and accuracy. We report per-subgroup values, macro-averages, and worst-group results. These choices mirror established practice and enable direct comparison to prior PEFT–fairness evaluations discussed in §3. Formal definitions and computation details appear in Appendix B.

### 4.2 PROTOCOL

We evaluate our methods on a main generative base model, LLaMA2-7B (Touvron et al., 2023)[1], and two compact baselines for CivilComments toxicity, DistilBERT (Sanh et al., 2019a)[2] and Qwen2.5-

---

[1] LLaMA 2 is licensed under the LLAMA 2 Community License, Copyright (c) Meta Platforms, Inc. All Rights Reserved. See `https://ai.meta.com/llama/license`.

[2] DistilBERT is released under the Apache 2.0 License. See `https://github.com/huggingface/transformers/blob/main/LICENSE`.

0.5B (Qwen Team, 2025)[3]. For the vision experiments on UTKFace, we use ViT-Base/16 (Dosovitskiy et al., 2021)[4], a 12-layer Vision Transformer with 768-dimensional hidden size and 16×16 patch embeddings. We select ViT-Base/16 as it represents a standard, high-capacity encoder architecture commonly used in fairness studies (Ding et al., 2024) and provides a strong counterpart to our text-based models, enabling cross-domain comparisons. Our fairness evaluations focus on two sensitive attributes: gender and race across both datasets, computing subgroup accuracy, DPD, and EOD. These selections span distinct architectures (decoder-only vs. encoder-only), parameter scales, tasks, and label taxonomies.

For FFT, the pretrained model was fine-tuned on the combined training data from all subgroups of the target attribute (gender or race). Evaluation was then performed on the test data from each corresponding subgroup, enabling fine-grained assessment of both performance and fairness. For LoRA, we followed the same training and evaluation procedure as FFT. The rank of LoRA's adaptation modules was set to 8, following Ding et al. (2024).

For task arithmetic, we applied a compositional fine-tuning approach. The training data was partitioned by subgroup (gender or race), and FFT was applied separately to each subgroup's data to produce fine-tuned models $\theta_i$. From these, we computed task vectors $\Delta\theta_i$ relative to the base model. These vectors were then merged using the approach described in Eq. (1), with a single, uniform scaling coefficient $\lambda$ applied to all vectors. $\lambda$ served as the sole hyperparameter in the merging process and was tuned on the training data. The evaluation metrics were computed in the same manner as for FFT and LoRA.

**Task vector coefficient adjustment.** Building on the task vector merging framework introduced in Eq. (1), we further explore the impact of the scaling coefficient $\lambda$ on fairness outcomes. Specifically, we vary the uniform task vector coefficient $\lambda$ across a broad range (from 0.0 to 1.0 with 0.1 intervals) and evaluate how this adjustment influences subgroup-level fairness metrics, including accuracy, DPD, and EOD.

**Impact of worst-performing subgroup task vectors on fairness and performance.** To investigate whether incorporating task vectors from underperforming subgroups can improve fairness without sacrificing overall performance, we first identified the lowest-performing subgroups within each attribute based on the average of DPD and EOD under the FFT setting. We excluded the "others" group from this analysis as it does not reflect the characteristics of any specific subgroup. This selection was informed by both our experimental results and those reported in Ding et al. (2024), which showed consistent patterns. For gender, the worst-performing subgroups were men and women; for race, they were Asian and Native American. We constructed a new model variant by injecting a worst-performing subgroup task vector into the base fine-tuned model:

$$\theta_{\text{new}} = \theta_{\text{SFT}} + \lambda(\theta_{\text{worst-performing subgroup}} - \theta_0),$$

where $\lambda$ controls the strength of the task vector injection. We varied $\lambda$ from 0.0 to 1.0 at 0.2 intervals to analyze the effect of this targeted addition on subgroup fairness metrics and overall accuracy.

## 5 RESULTS

### 5.1 THEORETICAL INTUITION.

To situate the subsequent analysis, we first provide a high-level rationale for why task vector scaling interacts systematically with group-wise fairness metrics. Scaling the task vectors modifies the model parameters in directions associated with each subgroup, and the resulting model $\theta(\boldsymbol{\lambda})$ can be viewed as a perturbation of the balanced reference point $\bar{\theta}$ obtained at $\lambda_g = 1$ for each $g$. Since group disparities are minimized at this balanced model, the behavior of fairness metrics is largely governed by the magnitude of the deviation $\theta(\boldsymbol{\lambda}) - \bar{\theta}$ and by the norms of the subgroup-specific task vectors that induce this deviation. For convenience, we write $\boldsymbol{\lambda} = (\lambda_g)_{g=1}^G$ for the vector of subgroup-specific scaling coefficients, with the balanced setting corresponding to $\lambda_g = 1$ for all $g$.

---

[3]See the Qwen2.5-0.5B model card and license details at `https://huggingface.co/Qwen/Qwen2.5-0.5B`.

[4]Original ViT implementation is released under the Apache 2.0 License. See `https://github.com/google-research/vision_transformer`.

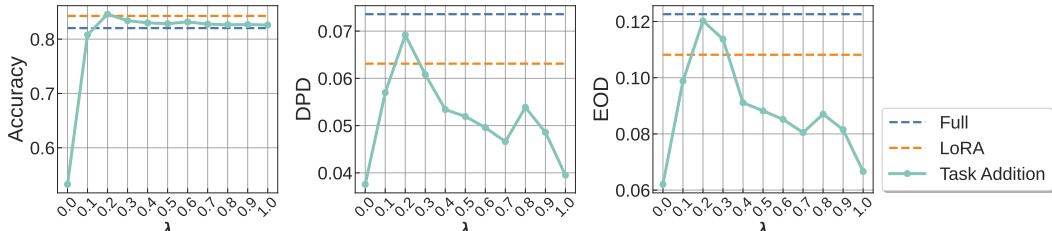

Figure 2: Varying the task arithmetic coefficient $\lambda$ and comparing against FFT (dashed baseline blue line) and LoRA (orange dashed) for macro-averaged accuracy (left), demographic parity difference (DPD, center), and equalized odds difference (EOD, right) on the **gender** subset. Higher accuracy is better; lower DPD/EOD indicate improved group fairness. For $\lambda \gtrsim 0.3$, task addition maintains *competitive accuracy* while *typically lowering* DPD/EOD relative to both baselines.

We complement our empirical findings with an analytical upper bound that links task vector scaling to fairness metrics. We make several assumptions for the analysis: **[A1]** the prediction score $p_\theta(x)$ is $L$-Lipschitz in $\theta$ and group-wise calibrated to its binarized prediction (see Appendix C), **[A2]** a task vector for each group $g$ (i.e. $\Delta\theta_g$) is obtained under an identical optimization protocol, **[A3]** the scaling coefficients satisfy the normalization $\sum_g \lambda_g = G$, **[A4]** the group distributions differ only through the sensitive attribute so that the balanced model $\bar{\theta} = \theta_0 + \frac{1}{G}\sum_g \Delta\theta_g$ satisfies $\mathrm{DPD}(\bar{\theta}) = 0$, **[A5]** near the binarization threshold, both $p_\theta(X)$ and $p_{\bar{\theta}}(X)$ have uniformly bounded class-conditional densities, i.e., for some $B_y > 0$ for each true label $y$ and sufficiently small $\gamma$.

Assumptions **[A1]** and **[A5]** impose mild regularity and calibration conditions on the prediction scores, whereas **[A2]**–**[A4]** correspond to comparatively natural requirements on the training setup. Under those assumptions, the DPD and EOD satisfy the following inequalities:

**Theorem (informal).** *Consider the merged model $\theta(\boldsymbol{\lambda}) = \theta_0 + \sum_g \lambda_g \Delta\theta_g$. Then DPD and EOD satisfy:*

$$\mathrm{DPD}(\theta(\boldsymbol{\lambda})) \leq U(\boldsymbol{\lambda}) = 2L \sum_g \big|\lambda_g - 1\big| \, \|\Delta\theta_g\|_2, \quad \text{for a Lipschitz constant } L,$$

$$\mathrm{EOD}(\theta(\boldsymbol{\lambda})) \leq \mathrm{EOD}(\bar{\theta}) + 4\sqrt{(B_0 + B_1)\,U(\boldsymbol{\lambda})}.$$

A full derivation and tighter constants are provided in Appendix C (Proposition 1,2).

These upper bounds explain the empirical trends observed in Figure 2. Both DPD and EOD share a common upper-bound term $U(\boldsymbol{\lambda})$, which monotonically decreases as $\lambda$ $(= \lambda_1 = \cdots = \lambda_G)$ approaches 1 for each $g$. Notably, the bound $\mathrm{DPD}(\theta(\boldsymbol{\lambda})) \leq U(\boldsymbol{\lambda})$ together with $U(\boldsymbol{\lambda}) \to 0$ as $\lambda_g \to 1$ implies that deviations of $\boldsymbol{\lambda}$ exert a pronounced influence on DPD. $B_0$ and $B_1$ are class-specific constants that upper-bound the score density near the decision threshold for y=0 and y=1, respectively. Intuitively, this bound shows that deviations of the scaling coefficient $\boldsymbol{\lambda}$ from the balanced setting ($\lambda_1 = \cdots = \lambda_G = 1$) enlarge disparities in proportion to the norms of subgroup task vectors.

## 5.2 EMPIRICAL RESULTS OVERVIEW.

**Key takeaway.** Across all datasets and architectures we study, task arithmetic exposes two complementary mechanisms for navigating fairness–accuracy trade-offs. First, sweeping a single scalar coefficient $\lambda$ over uniformly merged subgroup task vectors yields a smooth fairness–utility frontier: over a broad range of $\lambda$, task addition matches FFT/LoRA (or SFT/LoRA) accuracy while substantially reducing DPD and EOD on Berkeley D-Lab hate speech and *Civil Comments*, and in our additional vision experiments on UTKFace, across LLaMA-2, DistilBERT, Qwen-2.5, and ViT-Base. Second, injecting individual subgroup task vectors into an FFT model (Section 5.4) produces structured, subgroup-dependent shifts in fairness: some subgroup vectors (e.g., Men, Asian) move the model along favorable fairness–accuracy frontiers, while others (e.g., Native American) can worsen DPD/EOD even when accuracy remains stable. This behavior is consistent with the

group-weighted ERM view developed in Section 5.1 and Appendix C.2, where sensitivity to $\lambda$ scales with $\|\Delta\theta_g\|_2$. Together, these observations show that task arithmetic provides both a *global* control knob (a uniform $\lambda$) and *subgroup-targeted* knobs (the $\Delta\theta_g$) that are not exposed by standard FFT/LoRA training. The empirical trends and their first-order mechanism also align with prior literature: our macro-averaged accuracy, DPD, and EOD findings for FFT and LoRA are consistent with (Ding et al., 2024).

Figures 1a and 1b instantiate this story for hate-speech detection on Berkeley D-Lab with LLaMA-2 by comparing FFT, LoRA, and task addition across gender and race subgroups. For task addition, we selected $\lambda = 0.8$ for gender and $\lambda = 0.5$ for race, as these achieved the highest average training accuracy across three random seeds within the tested range $\lambda \in [0.0, 1.0]$. These visualizations provide a direct comparison of subgroup-wise model behavior. From the subgroup-level bar plots in Figure 1, we observe that accuracy remains consistently high and comparable across all three adaptation methods, regardless of subgroup. On *Civil Comments*, on both DistilBERT and Qwen-2.5, task addition similarly reduces group disparities while keeping accuracy competitive (see Appendix F and Table 5 for full confidence intervals and results).

We also observe that, relative to FFT, task addition improves fairness in five of seven gender subgroups and in three of eight race subgroups, with no single method dominating across all groups. The subgroup-dependent nature of these effects motivates treating $\lambda$ as a deliberate tuning knob and inspecting subgroup behavior explicitly. As shown in Appendix C.2, task addition realizes a group-weighted ERM in the linearized model: concretely, $\theta(\boldsymbol{\lambda}) = \theta_0 + \sum_g \lambda_g \Delta\theta_g$ coincides with the one-step minimizer of a first-order surrogate where subgroup $g$ is re-weighted by $\lambda_g$. This explains the smooth fairness–utility frontier traced by sweeping $\lambda$, and Theorem 5.1 predicts larger parity swings for groups with larger $\|\Delta\theta_g\|_2$. The observed curves in Fig. 2 align with those predictions without further assumptions.

### 5.3 CONTROLLING ACCURACY AND FAIRNESS METRICS THROUGH LAMBDA.

Figure 2 illustrates the overall performance of FFT, LoRA, and task arithmetic as the scaling coefficients for task addition vary from 0.0 to 1.0. We observe how varying the task-arithmetic coefficient $\lambda$ impacts macro-averaged accuracy (left), demographic parity difference (DPD, center), and equalized odds difference (EOD, right) on a gender subset of the data. As $\lambda$ increases from 0.0 to 0.2, we observe a peak in accuracy, but this configuration yields higher DPD and EOD, indicating reduced fairness. Beyond $\lambda = 0.3$, accuracy remains competitive compared to FFT and LoRA, while both DPD and EOD progressively decline, suggesting that fairness improves without severely compromising performance. Notably, these task addition curves stay consistently lower than FFT and LoRA in terms of DPD and EOD at higher $\lambda$ values. Overall, this ablation could indicate that tuning $\lambda$ provides a practical mechanism for balancing accuracy and fairness objectives, offering guidelines for practitioners who wish to fine-tune fairness outcomes while maintaining strong predictive performance.

### 5.4 SUBGROUP-TARGETED VECTORS: GAINS WITH TRADE-OFFS

To further analyze the effects of subgroup-specific task composition, Figure 3a–3b illustrate heatmaps where the y-axis lists each method or configuration under evaluation: FFT as baseline, followed by task arithmetic with varying scaling coefficients (0.0 to 1.0 with 0.2 intervals). The x-axis represents the subgroups— (e.g., Women, Trans, etc. for Gender). Each cell shows the corresponding performance metric (e.g., macro-averaged accuracy, DPD, or EOD for a given method on a specific subgroup. For these experiments, we added the task vector of the worst-performing subgroups (Women and Men for the gender dataset subset, and Asian, and Native American for the race dataset subset) to the FFT model, as explained earlier.

We generally observe that increasing the scaling coefficient $\lambda$ tends to improve overall accuracy, consistent with the trends observed in Figure 2. However, effects are not uniform across all subgroups. In the race-based plots, for example, the Asian subgroup consistently achieves the highest accuracy and lowest DPD/EOD—highlighting a recurring tradeoff where performance gains for one group may exacerbate disparities for others. When the Women task vector is added (Figure 3b), accuracy improves for the Trans Women subgroups. However, fairness metrics for subgroups such as Men tend to worsen as the scaling coefficient $\lambda$ increases.

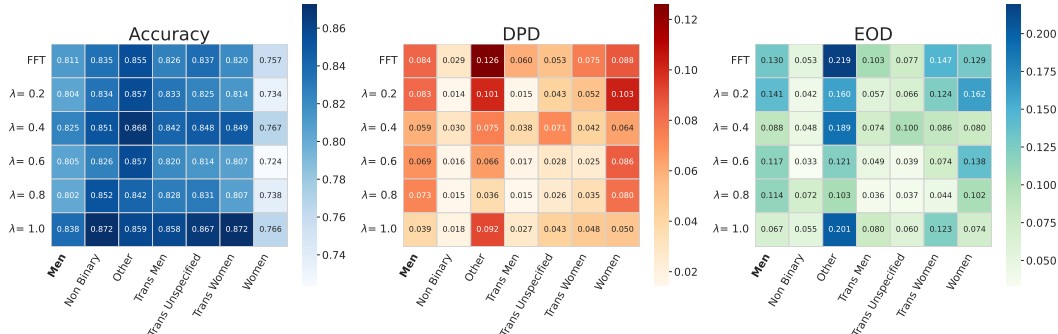

(a) When **Men** task vector added to the FFT model on the **gender** subset.

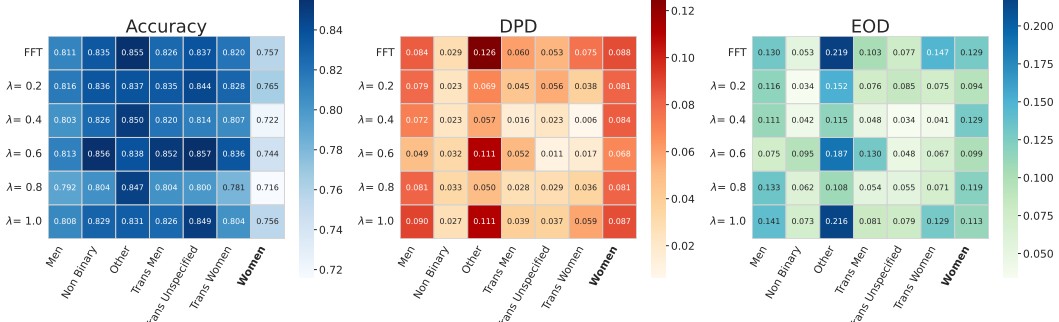

(b) When **Women** task vector added to the FFT model on the **gender** subset.

Figure 3: Heatmaps of Accuracy (left), DPD (center), and EOD (right) for Men (top) and Women (bottom) subgroups under the baseline FFT model ($\lambda = 0.0$) and with increasing $\lambda$ values from 0.2 to 1.0 in 0.2 increments. The task vector for Men was added on the gender subset (top), and the task vector for Women was added on the gender subset (bottom). Darker cells indicate higher values on each metric's scale; for DPD/EOD, lower values are better.

In Figure 3a, injecting the Men task vector improves performance for some subgroups, yet Women consistently show lower accuracy and do not see consistent fairness improvements at higher $\lambda$. Some groups (e.g., Other, Trans Men, Trans Women) begin with relatively poor fairness under FFT and show partial improvements with task vector addition. Still, these improvements are not universal—for example, the Other subgroup often retains high EOD values regardless of $\lambda$. Likewise, Native American accuracy remains mostly unchanged across $\lambda$, while fairness metrics can deteriorate when injecting task vectors for other groups. To visualize these results in more detail, Figure 4a shows macro-averaged accuracy, DPD, and EOD for the Men task vector added to the FFT model. The plots illustrate how varying the scaling coefficient $\lambda$ impacts overall performance and fairness, highlighting the effects of subgroup-specific task injection. We can observe in Figure 4a that injecting the Men task vector into the FFT model results in a slight accuracy gain and a clear monotonic decrease in both DPD and EOD as $\lambda$ increases—indicating a favorable and consistent improvement in fairness on the gender subset.

However, Figure 4b and the additional plots in Figures 10 and 11 in Appendix E.2 show more varied patterns as seen on Figures 3a and 3b. When injecting the Native American task vector (Figure 11), accuracy remains stable while fairness seems to decrease (increased DPD and EOD). Asian (Figure 10) shows the same behavior as injecting the Men task vector (Figure 4a), positive increase of fairness metrics as $\lambda$ increases. These results show that injecting task vectors shifts fairness and performance in a group-specific manner, tracing a clear fairness–utility frontier. This heterogeneity is expected: per §5.2 and Theorem 5.1, sensitivity scales with $|\Delta\theta_g|_2$. Practically, task vector merging thus offers a *subgroup-conditioned* control knob: identifying which $\Delta\theta_g$ help or hurt which groups provides a new actionable design consideration that SFT/LoRA do not expose, and that hasn't been explored in previous task arithmetic literature.

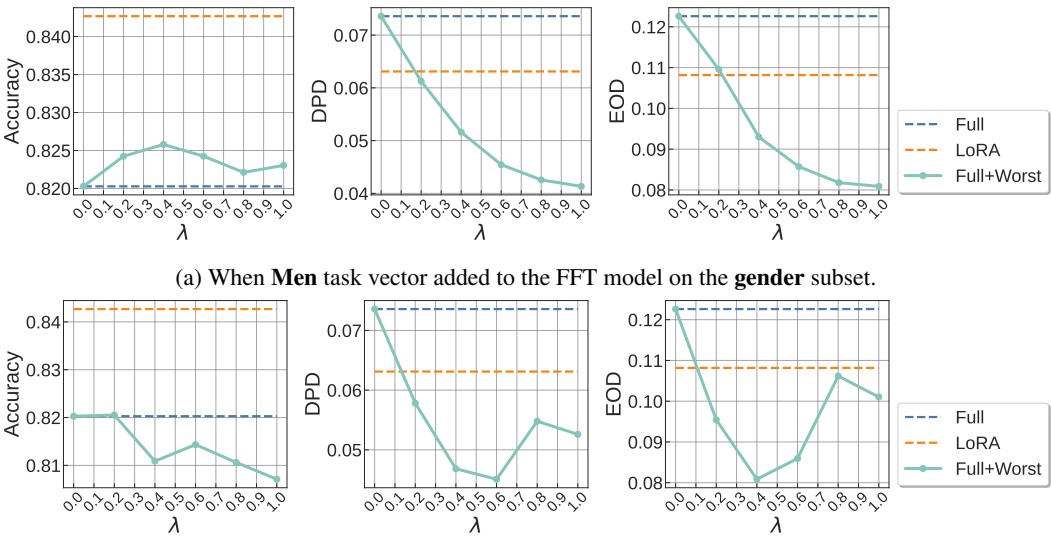

(a) When **Men** task vector added to the FFT model on the **gender** subset.

(b) When **Women** task vector added to the FFT model on the **gender** subset.

Figure 4: Impact of injecting the **Men (a)** and **Women (b)** subgroup task vectors into the FFT model on the gender data subset. The plot illustrates how scaling coefficient $\lambda$ reduces DPD and EOD, outperforming the baseline FFT (blue dashed) and LoRA (orange dashed), with negligible impact on macro-averaged accuracy.

# 6 CONCLUSION AND LIMITATIONS

**Conclusion.** In this study, we investigated how task arithmetic with task vectors affects group fairness, in comparison to conventional FFT and LoRA methods. We conducted experiments to assess how adding a fairness-oriented task vector impacts prediction accuracy and fairness metrics, including DPD and EOD across multiple subgroups. The results indicate that, with appropriate settings of the scalar coefficient $\lambda$, task arithmetic can improve DPD and EOD without significantly compromising overall model accuracy. Notably, low to moderate values of $\lambda$ often reduced prediction bias in minority groups compared to FFT and LoRA, a pattern we observe across three datasets (hate speech, toxicity, and age detection) and four model families (LLaMA-2, DistilBERT, Qwen-2.5, ViT-Base).

Furthermore, the task-arithmetic formulation exposes a single interpretable parameter $\lambda$, which allows practitioners to trace subgroup-specific changes along a fairness–accuracy trade-off curve. This transparency makes it easier to diagnose when edits disproportionately benefit or harm particular groups.

**Limitations.** Despite these promising results, several limitations remain. First, our experiments focus on the open-weight 0.5–7B regime (DistilBERT, Qwen-0.5B, LLaMA-2-7B, ViT-Base), where task arithmetic is practically deployable and comparable to prior work. This setting gives us full weight access to construct subgroup task vectors and compute exact DPD/EOD. We do not evaluate against proprietary frontier models; prior work shows that prompt-based debiasing can offer partial gains but is brittle and does not provide principled control over group-level disparities (Ma et al., 2023; Furniturewala et al., 2024; Chu et al., 2024). Extending our framework to larger or API-only models is an important direction for future work. Second, we study a simple scaling scheme based on a shared global coefficient $\lambda$. This keeps the intervention interpretable but cannot capture richer multidimensional fairness constraints. More expressive parameterizations (e.g., multiple coefficients or optimization over combinations of $\lambda$, represent natural extensions). Finally, our evaluation covers a limited set of classification tasks with single sensitive attributes. The effectiveness of task arithmetic may depend on dataset and subgroup structure, so assessing generalization to broader domains and intersectional settings remains important.

**Reproducibility statement.** We provide code [5], configs, and scripts to reproduce all experiments, including data preprocessing, training, and evaluation. All datasets and base models used are open-source/publicly available; we include scripts to fetch the exact versions. Exact hyperparameters, model identifiers, and implementation details are documented in the appendix, along with seeds and hardware/software specs. Results are reported over multiple runs, and we provide instructions to regenerate all figures and tables from logged outputs.

**Acknowledgments** Our deepest gratitude goes out to the anonymous reviewers whose invaluable insights substantially enhanced the quality of this manuscript. This work was supported by RBC Borealis through the RBC Borealis AI Global Fellowship Award, which was awarded to Hiroki Naganuma. We are also profoundly grateful to The Masason Foundation for their generous support in providing computational resources and for fostering an environment that encourages deep research collaboration. The computation resources were further supported by "TSUBAME Encouragement Program for Young/Female Users" of Center for Information Infrastructure at Institute of Science Tokyo and by "Joint Usage/Research Center for Interdisciplinary Large-scale Information Infrastructures" in Japan.

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

APPENDIX

# A  EXTENDED RELATED WORKS

**Task arithmetic: efficiency and interpretability.**  Task vectors offer a computationally efficient framework for editing and analyzing model behavior. Once a task vector is computed—namely, the weight difference between a base model and its fine-tuned variant (Ilharco et al., 2023; Zhang et al., 2024; Yoshida et al., 2025b)—no additional training data or retraining is required to transfer or remove task-specific capabilities. By treating each fine-tuning update as a direction in weight space, practitioners can combine or negate these updates through simple addition or subtraction (Ilharco et al., 2023). This modularity not only reduces computational overhead but also enhances interpretability by isolating the contribution of each task. Beyond modularity, task arithmetic can reveal valuable information about how and where a model adapts to new tasks. Li et al. (2024) show a near-linear relationship between data size and the norm of a task vector, suggesting that over-represented tasks can dominate weight space shifts in multi-task settings. In addition, the orientation of task vectors can indicate synergies or conflicts among tasks (Li et al., 2025), and decomposing these vectors by layer can pinpoint which parts of the model are most affected (Zhang et al., 2024; Gargiulo et al., 2025). Hence, task vectors offer a promising lens for diagnosing training dynamics and identifying potential biases.

**Vision–Language Models and Demographic Bias Audits.**  Audits of vision–language systems consistently report demographic biases in zero-shot classification, retrieval, and captioning, along with mixed effectiveness of post-hoc debiasing methods (Ali et al., 2023). Large studies of CLIP-style models show that scaling data or parameters does not reliably reduce these disparities, which vary with dataset composition and training regimes (Sahili et al., 2025). Even current vision–language models exhibit measurable group-level gaps under standard prompting (Wu et al., 2025).

# B  FAIRNESS METRICS

## B.1  DEMOGRAPHIC PARITY DIFFERENCE (DPD) (AGARWAL ET AL., 2018; 2019)

DPD measures how varied the model's rate of positive predictions is across attributes. This metric is calculated as follows:

$$M_{\text{DPD}} = \left| \Pr[f(X) = 1 \mid A = 1] - \Pr[f(X) = 1 \mid A = 0] \right|,$$

where $A$ is the sensitive attributes, $f(X)$ is the prediction from the models, and $X$ is the feature vector. The larger the DPD, the greater the difference in prediction outcomes across attributes, indicating greater unfairness in the model predictions.

## B.2  EQUALIZED ODDS DIFFERENCE (EOD) (DING ET AL., 2024)

EOD is a metric that measures whether the model exhibits similar predictive performance in terms of true and false positives, regardless of the attribute.

$$M_{\text{eod}} = \max \left\{ M_{\text{TP}}, M_{\text{FP}} \right\}. \tag{3}$$

Here, letting $Y$ denote the true label, $M_{TP}$ and $M_{FP}$ are defined as follows:

$$M_{\text{TP}} = \left| \Pr[f(X) = 1 \mid Y = 1, A = 1] - \Pr[f(X) = 1 \mid Y = 1, A = 0] \right|,$$

$$M_{\text{FP}} = \left| \Pr[f(X) = 1 \mid Y = 0, A = 1] - \Pr[f(X) = 1 \mid Y = 0, A = 0] \right|.$$

### B.3 ACCURACY PARITY

Accuracy parity refers to the expectation that a classifier achieves comparable accuracy across different sensitive attribute groups. Formally, accuracy parity is satisfied when the probability of correct classification is equal across groups, i.e.,

$$\mathbb{E}(Y = \hat{Y} \mid S = 0) = \mathbb{E}(Y = \hat{Y} \mid S = 1), \tag{4}$$

This notion of fairness ensures that all subgroups receive equally reliable predictions, and is particularly relevant in applications where consistent model performance across demographics is critical. Unlike statistical parity or equal opportunity, accuracy parity focuses on equal overall correctness rather than specific error types or outcome rates (Quan et al., 2023).

We observed **high degree of accuracy parity** in both gender and race settings, as the accuracy differences between subgroups are negligible, indicating that the model performs consistently across all groups.

## C UPPER BOUNDS ON DPD AND EOD FOR OPTIMAL TASK–VECTOR SCALING

### C.1 NOTATION AND ASSUMPTIONS

**A1 Smooth and calibrated scores.** Soft scores $p_\theta(x)$ are $L$-Lipschitz in the parameters, i.e.,
$$|p_\theta(x) - p_{\theta'}(x)| \le L \, \|\theta - \theta'\|_2 \quad \text{for all } x.$$
Hard predictions are obtained by thresholding a group-agnostic score,
$$f_\theta(x) = 1\{p_\theta(x) \ge t\}.$$
Moreover, soft scores and hard predictions agree in expectation at the group level: for every group $g$,
$$\mathbb{E}\big[p_\theta(X) \mid A = g\big] \;=\; \Pr\big(f_\theta(X) = 1 \mid A = g\big).$$

**A2 Task vectors.** For each group $g \in \{1, \dots, G\}$, $\Delta\theta_g := \theta_0^{(g)} - \theta_0$ is obtained with the *same* learning rate and schedule.

**A3 Scaling coefficients.** Coefficients obey $\sum_g \lambda_g = G$.

**A4 Balanced data assumption.** The joint distribution satisfies $\mathcal{D} = \bigcup_g \mathcal{D}_g$ where all $\mathcal{D}_g$ share the same conditional distribution except for the sensitive attribute label.

**A5 Local uniform margin condition near the threshold.** For each class label $y \in \{0, 1\}$, there exists a constant $B_y > 0$ and a neighborhood $\mathcal{N}(\bar\theta)$ of $\bar\theta$ in parameter space such that, for all $\theta \in \mathcal{N}(\bar\theta)$, all groups $g$, and all sufficiently small $0 < \gamma \le \gamma_0$,
$$\Pr\left(|p_\theta(X) - t| \le \gamma \mid Y = y, A = g\right) \le B_y \gamma,$$
$$\Pr\left(|p_{\bar\theta}(X) - t| \le \gamma \mid Y = y, A = g\right) \le B_y \gamma.$$
In words: around the decision threshold, the class-conditional score densities of both $\theta$ and $\bar\theta$ are uniformly bounded by $B_y$ in a neighborhood of $\bar\theta$. This is the standard "low density around the decision boundary" or "margin" assumption, here required to hold uniformly in a local neighborhood.

The merged model is
$$\theta(\boldsymbol{\lambda}) \;=\; \theta_0 + \sum_g \lambda_g \, \Delta\theta_g.$$

By [A1], the soft scores and hard predictions have matching expectations, so DPD computed from $p_\theta$ coincides with DPD based on the classifier $f_\theta$ in Appendix B. In particular,
$$\mathrm{DPD}(\theta) \;=\; \Big| \mathbb{E}_{\mathcal{D}_1}[p_\theta] - \mathbb{E}_{\mathcal{D}_0}[p_\theta] \Big|.$$

## C.2  TASK ADDITION AND WEIGHTED ERM

**Lemma 1** (First-order link). *Let $\ell(\theta; x)$ be the training loss. For any non-negative $\{\lambda_g\}$,*

$$\theta(\boldsymbol{\lambda}) \approx \arg\min_{\theta} \sum_g \lambda_g \, \mathbb{E}_{x \sim \mathcal{D}_g} \big[ \ell(\theta_0; x)$$
$$+ \nabla_\theta \ell(\theta_0; x)^\top (\theta - \theta_0) \big].$$

*That is, task addition gives the* first-order *solution of a group-weighted ERM.*

*Proof.* Insert the linear Taylor expansion of $\ell$ at $\theta_0$ and minimise the resulting quadratic form; the solution is exactly $\theta(\boldsymbol{\lambda})$. $\square$

**Implication.**  Hence, deviations $|\lambda_g - 1|$ alter the group weights and therefore *directly pushes DPD upward*, as made explicit in Proposition 1 below.

## C.3  DPD UPPER BOUND

**Proposition 1** (DPD bound). *Under Assumptions **A1**–**A4**,*

$$\mathrm{DPD}\big(\theta(\boldsymbol{\lambda})\big) \leq 2L \sum_g |\lambda_g - 1| \, \|\Delta\theta_g\|_2.$$

*Proof.* Define $\bar{\theta} := \theta_0 + \frac{1}{G} \sum_g \Delta\theta_g$. Assumption **A4** gives $\mathrm{DPD}(\bar{\theta}) = 0$. Define $h(x) := p_{\theta(\boldsymbol{\lambda})}(x) - p_{\bar{\theta}}(x)$. Then $\mathrm{DPD}(\theta(\boldsymbol{\lambda})) = |\mathbb{E}_{\mathcal{D}_1}[h] - \mathbb{E}_{\mathcal{D}_0}[h]|$. Triangle and Jensen inequality yield $\leq 2L \|\theta(\boldsymbol{\lambda}) - \bar{\theta}\|_2$. Finally, $\theta(\boldsymbol{\lambda}) - \bar{\theta} = \sum_g (\lambda_g - 1)\Delta\theta_g$ and the triangle inequality give the stated bound. $\square$

## C.4  POINTWISE SYMMETRIC BOUND VIA A RAMP FUNCTION

**Lemma 2.**

$$\left| \mathbf{1}\{p_\theta(x) \geq t\} - \mathbf{1}\{p_{\theta'}(x) \geq t\} \right| \leq \frac{L}{\gamma} \|\theta - \theta'\|_2 + \mathbf{1}\{|p_\theta(x) - t| \leq \gamma\} + \mathbf{1}\{|p_{\theta'}(x) - t| \leq \gamma\}.$$

*Proof.* Define the ramp function $\phi_{t,\gamma} : \mathbb{R} \to [0, 1]$ by

$$\phi_{t,\gamma}(u) := \begin{cases} 0, & u \leq t - \gamma, \\ \dfrac{u - (t - \gamma)}{\gamma}, & t - \gamma < u < t, \\ 1, & u \geq t. \end{cases} \tag{5}$$

This function is $1/\gamma$–Lipschitz and satisfies the following inequalities: $|\mathbf{1}\{u \geq t\} - \phi_{t,\gamma}(u)| \leq \mathbf{1}\{|u - t| \leq \gamma\}$.

For arbitrary $\theta, \theta'$, applying the triangle inequality gives

$$\left| \mathbf{1}\{p_\theta(x) \geq t\} - \mathbf{1}\{p_{\theta'}(x) \geq t\} \right| \leq T_1 + T_2 + T_3, \tag{6}$$

where

$$T_1 = \left| \mathbf{1}\{p_\theta(x) \geq t\} - \phi_{t,\gamma}(p_\theta(x)) \right| \tag{7}$$
$$T_2 = \left| \phi_{t,\gamma}(p_\theta(x)) - \phi_{t,\gamma}(p_{\theta'}(x)) \right| \tag{8}$$
$$T_3 = \left| \phi_{t,\gamma}(p_{\theta'}(x)) - \mathbf{1}\{p_{\theta'}(x) \geq t\} \right|. \tag{9}$$

By the ramp construction,

$$T_1 \leq \mathbf{1}\{|p_\theta(x) - t| \leq \gamma\}, \quad T_3 \leq \mathbf{1}\{|p_{\theta'}(x) - t| \leq \gamma\}. \tag{10}$$

By the Lipschitzness of $\phi_{t,\gamma}$ and the soft scores,

$$T_2 \leq \frac{1}{\gamma} |p_\theta(x) - p_{\theta'}(x)| \leq \frac{L}{\gamma} \|\theta - \bar{\theta}\|_2. \tag{11}$$

Thus we obtain the stated equation. $\square$

## C.5 EOD UPPER BOUND

**Proposition 2** (EOD bound). *Under Assumptions **A1**–**A5**,*

$$\text{EOD}(\theta(\boldsymbol{\lambda})) \le \text{EOD}(\bar{\theta}) + 4\sqrt{(B_0 + B_1)\, U(\boldsymbol{\lambda})},$$

*where $U(\boldsymbol{\lambda})$ is the upper bound of EOD:*

$$U(\boldsymbol{\lambda}) = 2L \sum_g |\lambda_g - 1|\|\Delta\theta_g\|_2.$$

*Proof.*

$$\text{EOD}(\theta) = \max_{y\in\{0,1\}}\ \max_{g,g'}\Big|\Pr[f_\theta(X)=1\mid Y=y, A=g] - \Pr[f_\theta(X)=1\mid Y=y, A=g']\Big|. \tag{12}$$

$$\begin{aligned}
\Big|&\Pr[f_\theta = 1\mid Y=y, A=g] - \Pr[f_\theta = 1\mid Y=y, A=g']\Big| \\
&\le \Big|\Pr[f_\theta = 1\mid Y=y, A=g] - \Pr[f_{\bar\theta} = 1\mid Y=y, A=g]\Big| \\
&\quad + \Big|\Pr[f_{\bar\theta} = 1\mid Y=y, A=g] - \Pr[f_{\bar\theta} = 1\mid Y=y, A=g']\Big| \\
&\quad + \Big|\Pr[f_{\bar\theta} = 1\mid Y=y, A=g'] - \Pr[f_\theta = 1\mid Y=y, A=g']\Big|.
\end{aligned} \tag{13}$$

Put

$$m_{y,g}^{(\theta)}(\gamma) := \Pr(|p_\theta(X) - t| \le \gamma \mid Y=y, A=g) \tag{14}$$

$$m_{y,g}^{(\bar\theta)}(\gamma) := \Pr(|p_{\bar\theta}(X) - t| \le \gamma \mid Y=y, A=g). \tag{15}$$

By Lemma 2

$$\Big|\Pr[f_\theta = 1\mid Y=y, A=g] - \Pr[f_{\bar\theta} = 1\mid Y=y, A=g]\Big| \le \frac{L}{\gamma}\|\theta - \bar\theta\|_2 + m_{y,g}^{(\theta)}(\gamma) + m_{y,g}^{(\bar\theta)}(\gamma). \tag{3}$$

Now take the maximum over $g, g'$ and $y \in \{0,1\}$. By the definition of EOD, we get

$$\text{EOD}(\theta) \le \text{EOD}(\bar\theta) + \frac{2L}{\gamma}\|\theta - \bar\theta\|_2 + 2\sum_{y\in\{0,1\}}\big(m_y^{(\theta)}(\gamma) + m_y^{(\bar\theta)}(\gamma)\big). \tag{16}$$

From assumption **A5**,

$$\sum_{y\in\{0,1\}}\big(m_y^{(\theta)}(\gamma) + m_y^{(\bar\theta)}(\gamma)\big) \le 2(B_0 + B_1)\gamma. \tag{17}$$

Substituting this into equation 16 yields

$$\text{EOD}(\theta) \le \text{EOD}(\bar\theta) + \frac{2L}{\gamma}\|\theta - \bar\theta\|_2 + 4(B_0 + B_1)\gamma. \tag{18}$$

Then we minimize the upper bound over $\gamma > 0$. By the arithmetic-geometric mean inequality, $F(\gamma) := \frac{2L}{\gamma}\|\theta - \bar\theta\|_2 + 4(B_0 + B_1)\gamma$ is minimized when $\gamma = \sqrt{\frac{L}{2(B_0+B_1)}}\|\theta - \bar\theta\|_2$:

$$\min_\gamma F(\gamma) = 4\sqrt{2L(B_0 + B_1)\,\|\theta - \bar\theta\|_2}. \tag{19}$$

Then, the triangle inequality gives the stated bound:

$$\text{EOD}(\theta) \leq \text{EOD}(\bar{\theta}) + 4\sqrt{2L(B_0 + B_1)\,\|\theta - \bar{\theta}\|_2} \qquad (20)$$

$$= \text{EOD}(\bar{\theta}) + 4\sqrt{(B_0 + B_1)\,U(\boldsymbol{\lambda})}. \qquad (21)$$

$\square$

## D  EXPERIMENTAL DETAILS

### D.1  COMPUTATIONAL RESOURCES AND SOFTWARE ENVIRONMENT

**Hardware and Software:**  All experiments presented in this study were performed using computational resources equipped with two NVIDIA H100 GPUs.  The experiments leveraged a GPU environment consisting of CUDA 12.1.0, cuDNN 9.0.0, and NCCL 2.20.5.

The experiments were conducted using Python 3.9.18, incorporating several essential Python libraries specifically optimized for deep learning tasks. The primary libraries included PyTorch (version 2.6.0), transformers (version 4.49.0), tokenizers (version 0.21.1), DeepSpeed (version 0.16.4), and Accelerate (version 1.5.2).

The training experiments utilized the DeepSpeed framework with the following key configurations: a gradient accumulation step of 4, optimizer offloaded to the CPU, zero redundancy optimizer at stage 2 (ZeRO-2), and mixed precision training employing FP16 and BF16 for enhanced performance and memory efficiency. All experiments were conducted with a total computational cost of approximately 30 GPU-hours.

**Protocol:**  We fine-tuned models based on the Llama-7B (Touvron et al., 2023) architecture obtained via HuggingFace repositories.  Each model was trained for 4 epochs, employing a cosine learning rate scheduler with a learning rate of $1 \times 10^{-5}$, a warm-up ratio of 0.01, and a weight decay of 0.001. Training utilized a per-device batch size of 2, with an effective batch size of 16 achieved through gradient accumulation. Reproducibility was ensured by setting a random seed of 13, 14, 15 across all experiments.

For Qwen2.5 experiments, models were trained for 2 epochs using a learning rate of $2 \times 10^{-5}$, a batch size of 16, and a sample fraction of 25% of the Civil Comments dataset. DistilBERT experiments utilized 2 epochs with a learning rate of $1 \times 10^{-5}$, a batch size of 16, and the full dataset (100% sample fraction). Both architectures employed a weight decay of 0.01 and evaluation/save strategies set to "epoch" with early stopping enabled.

For Low-Rank Adaptation (LoRA) experiments were conducted with a rank (lora_r) of 8, scaling factor (lora_alpha) of 16, and no dropout.

### D.2  DATASET

We use the Berkeley D-Lab hatespeech detection dataset (Kennedy et al., 2020b) [6] for our experiments.

The dataset is divided into subgroups based on the following attributes: *Race or Ethnicity*, *Religion*, *National Origin or Citizenship Status*, *Gender Identity*, *Sexual Orientation*, *Age*, and *Disability Status*. In our study, we use some of these subgroups to evaluate fairness.

Following Das et al. (2024), we binarize the hate speech score associated with each review using a threshold of 0.5 to determine whether the review constitutes hate speech. When multiple annotations exist for the same instance, we obtain one human annotation to avoid duplication.

## E  ADDITIONAL RESULTS

Here, we present results focusing on diverse subgroups, which we could not include in the main paper due to space constraints.

---

[6]`https://huggingface.co/datasets/ucberkeley-dlab/measuring-hate-speech`

### E.1  COMPARISON OF FFT, LORA, AND TASK ARITHMETIC

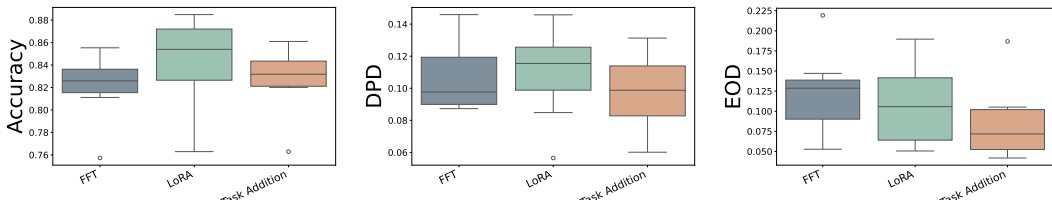

Figure 5: Boxplots of group-wise accuracy, demographic parity difference (DPD), and equalized odds difference (EOD) for FFT, LoRA, and task addition with coefficient ($\lambda = 0.8$) evaluated on the **gender** subset of the data. Higher accuracy is desirable, whereas lower DPD and EOD values indicate improved fairness. Boxplots show medians, interquartile ranges, and variability (with standard error across three seeds). While accuracy is similar across methods, Task Addition generally yields lower DPD and EOD medians than FFT and LoRA, suggesting a better balance between performance and fairness, though overlapping distributions imply these differences are not uniformly significant.

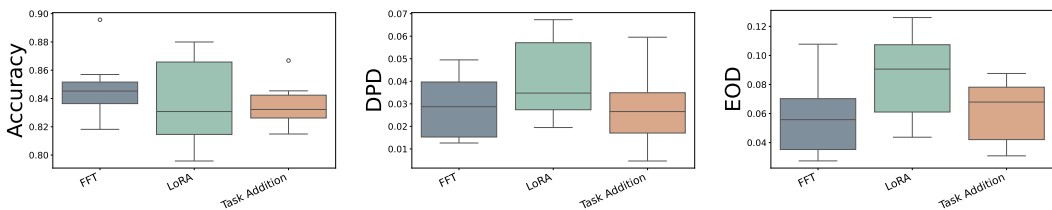

Figure 6: Boxplots of group-wise accuracy, demographic parity difference (DPD), and equalized odds difference (EOD) for —FFT, LoRA, and Task Addition with optimal coefficient ($\lambda = 0.5$) —evaluated on the **race** subset of the data. Higher accuracy is desirable, whereas lower DPD and EOD values indicate improved fairness. Boxplots show medians, interquartile ranges, and variability (with standard error across three seeds).

Figure 7 illustrates the overall performance of FFT, LoRA, and task arithmetic as the scaling for task arithmetic vary from 0.0 to 1.0. Trends observed reinforced results on the gender subset on Figure 2. Overall, $\lambda$ provides a practical mechanism for balancing accuracy and fairness objectives, and similarly there is a peak at $\lambda = 0.2$ for highest accuracy, and higher DPD and EOD (less fairness).

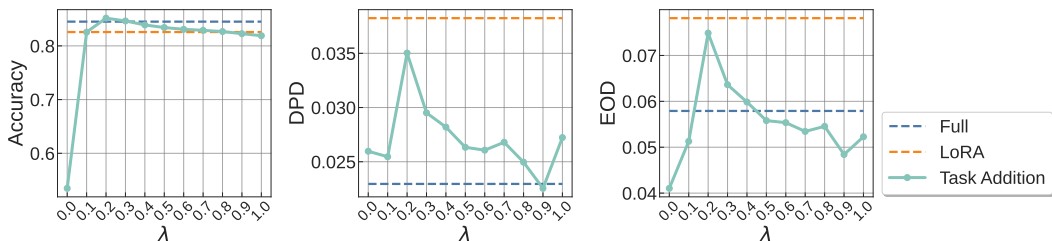

Figure 7: On a **race-focused** subset, we vary task arithmetic's coefficient $\lambda$ and compare it against FFT (purple dashed) and LoRA (orange dashed). The plots show group-wise accuracy (left), demographic parity difference (DPD, center), and equalized odds difference (EOD, right). Higher accuracy is better, while lower DPD and EOD indicate improved fairness. As $\lambda$ changes, task arithmetic remains competitive in accuracy and can reduce fairness gaps relative to the baselines.

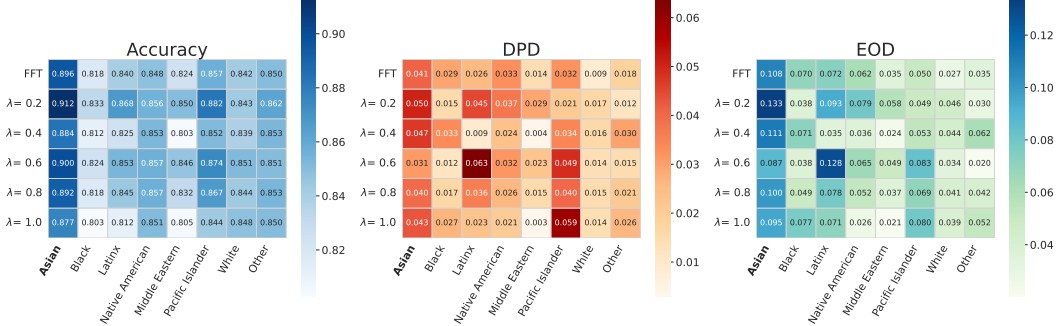

Figure 8: The task vector corresponding to **Asian** was added to the FFT model on the race data subset. Heatmap of Accuracy (left), DPD (center), and EOD (right) under the baseline (FFT) and increasing $\lambda$ values (0.2 to 1.0). Darker cells indicate higher values in each metric's scale; for DPD/EOD, lower is better.

## E.2 SUBGROUP-SPECIFIC TASK ADDITION TO FFT

We include additional heatmaps that visualize subgroup-wise performance across FFT and varying scaling coefficients for the FFT model injected with a worst-performing subgroup. These supplementary plots, which follow the same setup described earlier, are consistent with the trends observed in Figures 3a–3b.

In both gender and race subgroup experiments, increasing the scaling coefficient $\lambda$ generally leads to improved macro-averaged accuracy. However, its impact on fairness metrics—DPD and EOD—is less predictable and varies across subgroups. For instance, some subgroups benefit from improved fairness as their corresponding task vectors are added, while others experience increased disparity, even if accuracy remains stable or improves.

This nuanced behavior reflects a broader pattern: gains in performance for certain subgroups can sometimes come at the expense of fairness for others. Injecting task vectors from worst-performing subgroups does not consistently reduce disparities and, in some cases, can amplify them.

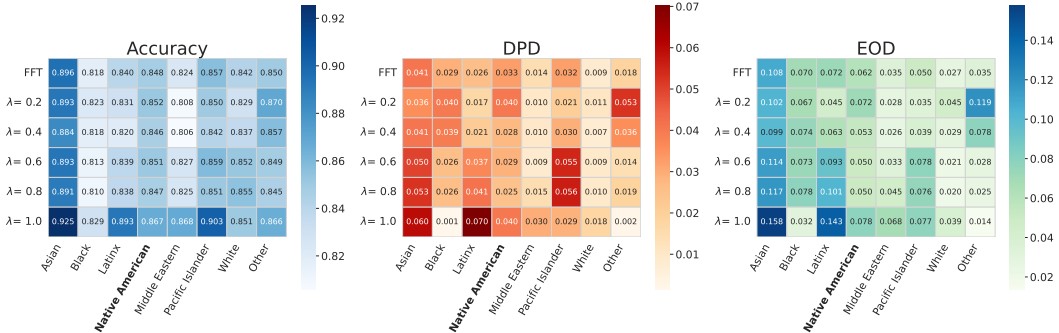

Figure 9: The task vector corresponding to **Native American** was added to the FFT model on the race data subset. Heatmap of Accuracy (left), DPD (center), and EOD (right) under the baseline (FFT) and increasing $\lambda$ values (0.2 to 1.0). Darker cells indicate higher values in each metric's scale; for DPD/EOD, lower is better.

Figures 10-11 and 4a-4b present additional results for the Full and Worst configuration, in which task vectors from the worst-performing subgroups (Native American, Asian, Men, and Women) are added to the FFT model. These plots show macro-averaged accuracy, DPD, and EOD as a function of the scaling coefficient $\lambda$.

Across these figures, we observe mixed effects: while accuracy generally remains stable or improves slightly, fairness outcomes vary by subgroup. In Figure 11, DPD and EOD worsen despite minimal accuracy changes. Meanwhile, Figure 4b reveals stable performance with minor fairness

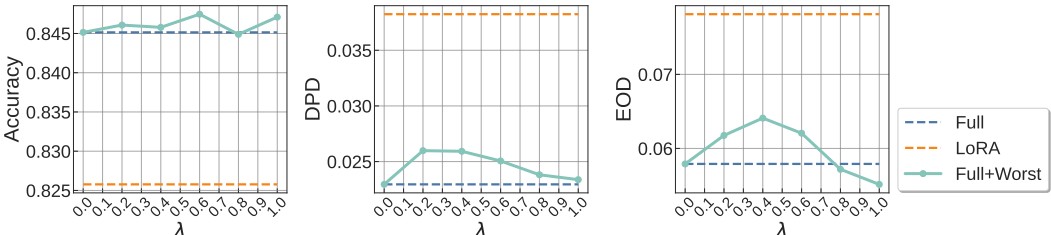

Figure 10: Effect of adding the **Asian** task vector to the FFT model on the **race** subset. Accuracy keeps competitive with increasing $\lambda$, and both DPD and EOD decrease consistently.

improvements, though gains are not consistent across metrics. These results further emphasize that task vector injection alone does not ensure universal fairness improvements and often introduces subgroup-specific trade-offs.

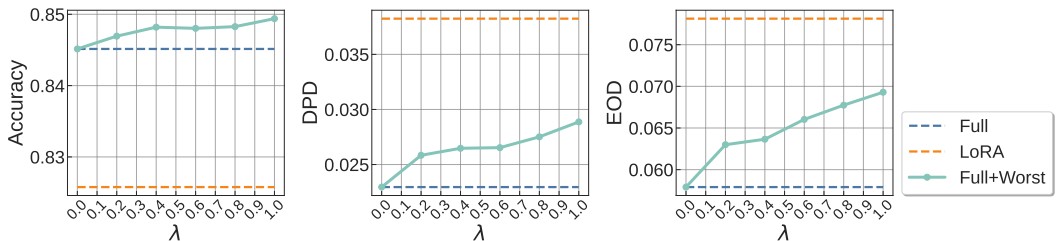

Figure 11: Results of injecting the **Native American** task vector into the FFT model. Accuracy shows minimal change across $\lambda$, while DPD and EOD increase (worsen fairness).

## F  ADDITIONAL EXPERIMENTS ON CIVIL COMMENTS

**Protocol & uncertainty.** Unless noted, we follow the LLaMA-2 setup (Section 4.2): SFT and LoRA ($r=8$) to obtain subgroup-specific models, compute task vectors w.r.t. the pretrained base, and merge with a uniform scalar $\lambda$. We sweep $\lambda$ on the validation split (maximize overall accuracy) and evaluate on the test split. Uncertainty is 95% stratified bootstrap over the test set (2,000 resamples, preserving group $\times$ label frequencies). When multiple seeds are used, we pool predictions before resampling. For accuracy, we additionally report Wilson CIs when relevant.

**At a glance.** On Civil Comments with DistilBERT (67M), task addition maintains accuracy within $\sim$0.6–1.1pp of SFT/LoRA while reducing fairness gaps: for *gender*, DPD drops by $\approx$41–54% and EOD by $\approx$34–47%; for *race*, DPD drops by $\approx$41–58% and EOD by $\approx$58–73% (midpoint comparisons). These patterns align with LLaMA-2 on the Berkeley D-Lab dataset (Table 5). As a complementary cross-architecture check, Qwen-2.5-0.5B on *gender* exhibits the same qualitative $\lambda$-controlled trade-off, improving substantially over LoRA with competitive accuracy.

### F.1  CIVIL COMMENTS — GENDER

**Notes.** Relative to LoRA, Qwen-2.5-0.5B task addition halves DPD/EOD ($\sim$54–56%) while regaining $\sim$3.3pp accuracy; relative to SFT, accuracy is lower and fairness is mixed (DPD comparable; EOD higher). DistilBERT shows consistent reductions in DPD/EOD with $\lesssim$1pp accuracy cost.

### F.2  CIVIL COMMENTS — RACE

**Discussion.** Together with LLaMA-2 on Berkeley D-Lab (Table 5), these experiments indicate that the $\lambda$-controlled fairness–utility trade-off extends across architectures and datasets: task addition typically preserves accuracy within $\sim$1pp while materially reducing worst-case DPD/EOD.

Table 2: **Civil Comments (Gender).** Headline metrics (Accuracy ↑, worst-case DPD ↓, worst-case EOD ↓). Entries are 95% CIs from stratified bootstrap

| Model/Method | Accuracy | Worst-DPD | Worst-EOD |
|---|---|---|---|
| DistilBERT  SFT | 0.9457–0.9476 | 0.0887–0.1101 | 0.6157–0.6433 |
| DistilBERT  LoRA | 0.9447–0.9453 | 0.0735–0.0812 | 0.5024–0.5084 |
| **DistilBERT  Task Addition** | **0.9395**[†] | **0.0454**[†] | **0.3358**[†] |
| Qwen-2.5-0.5B  SFT | 0.884–0.886 | 0.093–0.119 | 0.060–0.084 |
| Qwen-2.5-0.5B  LoRA | 0.774–0.790 | 0.210–0.251 | 0.232–0.362 |
| **Qwen-2.5-0.5B  Task Addition** | **0.810–0.820** | **0.100–0.103** | **0.130–0.143** |

[†] Point estimates.

Table 3: **Civil Comments (Race).** Headline metrics (Accuracy ↑, worst-case DPD ↓, worst-case EOD ↓). Models evaluated for this attribute are shown. CIs are 95% stratified bootstrap; [†] indicates point estimates.

| Model/Method | Accuracy | Worst-DPD | Worst-EOD |
|---|---|---|---|
| DistilBERT  SFT | 0.9467–0.9473 | 0.0987–0.0995 | 0.2568–0.3544 |
| DistilBERT  LoRA | 0.9446–0.9453 | 0.1360–0.1425 | 0.4649–0.4895 |
| **DistilBERT  Task Addition** | **0.9362**[†] | **0.0580**[†] | **0.1289**[†] |

# G  ADDITIONAL EXPERIMENTS ON UTKFACE (VIT-BASE/16)

**Protocol & uncertainty.**  We follow the same task vector protocol as in the NLP experiments: for each sensitive attribute (race or gender), we obtain subgroup-specific SFT and LoRA ($r=8$) models, compute task vectors relative to the ViT-Base/16 pretrained backbone, and merge them using a uniform scalar $\lambda$. Images are preprocessed following standard ViT practice (resize to $224^2$, patchify to $16 \times 16$, normalize using ImageNet statistics). The UTKFace age label is binarized following prior work: ages below a threshold form the "younger" class and ages above form the "older" class. We sweep $\lambda$ over the validation split (maximizing overall accuracy) and evaluate on the held-out test split. Uncertainty reflects 95% stratified bootstrap (2,000 resamples preserving group $\times$ label frequencies).

**At a glance.**  Across both *race* and *gender*, task addition again provides a favorable fairness–utility compromise, closely mirroring the patterns observed in NLP.

For **race**, SFT achieves the highest accuracy (0.8463–0.8599) but also the largest gaps (worst-DPD 0.3374–0.3881; worst-EOD 0.1815–0.2693). LoRA substantially reduces these disparities (DPD 0.2298–0.2789; EOD 0.1670–0.2238) but at the cost of a sharp accuracy drop (0.6389–0.6572). Task addition recovers much of this lost performance (0.7227–0.7384) while keeping worst-DPD/EOD close to LoRA and far below SFT (DPD 0.2390–0.2924; EOD 0.1700–0.2337), producing a more balanced trade-off.

For **gender**, we see the same trend. SFT again yields the best accuracy (0.8466–0.8600) but relatively large gaps (DPD 0.2513–0.2872; EOD 0.1446–0.1935). LoRA lowers DPD slightly but worsens EOD and reduces accuracy substantially (0.6389–0.6572). Task addition strikes the strongest balance: accuracy improves meaningfully over LoRA (0.7227–0.7384) while worst-DPD and worst-EOD are simultaneously reduced (DPD 0.1692–0.2080; EOD 0.0931–0.1379).

Overall, UTKFace confirms that task addition reproduces the same fairness–utility behavior observed on LLaMA-2 and Civil Comments—closing a large portion of LoRA's accuracy gap while preserving most of its fairness gains. This suggests the effect is consistent across domains (vision vs. text), model classes (ViT vs. transformers), and sensitive attributes (race vs. gender).

Table 4: **UTKFace (Race).** Binary age classification with ViT-Base/16. Accuracy ↑, worst-case DPD ↓, worst-case EOD ↓. Entries are 95% bootstrap CIs.

| Method | Accuracy | Worst-DPD | Worst-EOD |
|---|---|---|---|
| SFT | 0.8463–0.8599 | 0.3374–0.3881 | 0.1815–0.2693 |
| LoRA | 0.6389–0.6572 | 0.2298–0.2789 | 0.1670–0.2238 |
| **Task Addition** | **0.7227–0.7384** | **0.2390–0.2924** | **0.1700–0.2337** |

| Model | Race (95% CI) | Accuracy | Worst DPD | Worst EOD |
|---|---|---|---|---|
| LLaMA2-7B | SFT | 0.7901–0.9039 | 0.0000–0.0345 | 0.0000–0.0730 |
| | LoRA | 0.7599–0.9143 | 0.0000–0.0459 | 0.0000–0.1087 |
| | Task addition | **0.7972–0.8724** | **0.0000–0.0265** | **0.0000–0.1308** |
| DistilBERT | SFT | 0.9467–0.9473 | 0.0987–0.0995 | 0.2568–0.3544 |
| | LoRA | 0.9446–0.9453 | 0.1360–0.1425 | 0.4649–0.4895 |
| | Task addition | **0.9362** | **0.0580** | **0.1289** |

| Model | Gender (95% CI) | Accuracy | Worst DPD | Worst EOD |
|---|---|---|---|---|
| LLaMA2-7B | SFT | 0.7914–0.8491 | 0.0621–0.1125 | 0.0000–0.1794 |
| | LoRA | 0.8031–0.8823 | 0.0535–0.0596 | 0.0105–0.0906 |
| | Task addition | **0.8031–0.8823** | **0.0259–0.0943** | **0.0000–0.0858** |
| DistilBERT | SFT | 0.9457–0.9476 | 0.0887–0.1101 | 0.6157–0.6433 |
| | LoRA | 0.9447–0.9453 | 0.0735–0.0812 | 0.5024–0.5084 |
| | Task addition | **0.9395** | **0.0454** | **0.3358** |
| Qwen-2.5-0.5B | SFT | 0.884–0.886 | 0.093–0.119 | 0.060–0.084 |
| Qwen-2.5-0.5B | LoRA | 0.774–0.790 | 0.210–0.251 | 0.232–0.362 |
| **Qwen-2.5-0.5B** | **Task addition** | **0.810–0.820** | **0.100–0.103** | **0.130–0.143** |

Table 5: 95% confidence intervals. Models evaluated for each attribute are shown: LLaMA2-7B on Berkeley D-Lab; DistilBERT and Qwen-2.5-0.5B on Civil Comments (Qwen-2.5 for gender). Task addition maintains accuracy while showing competitive or improved fairness compared to SFT and LoRA.

## H  USE OF LARGE LANGUAGE MODELS (LLMS)

**Scope of assistance.** For polishing grammar, wording, concision, and transitions in the abstract, introduction, and discussion. Light edits on figure/table captions and section headings. And style normalization, enforcing consistent terminology and tense across sections. No ideas, claims, analyses, datasets, model architectures, experiments, or results originated from an LLM.

**Models and interface.** Edits were produced with state-of-the-art LLMs (e.g., ChatGPT/GPT-class models) via a standard chat interface. To preserve anonymity, no identifying information (author names, affiliations, or URLs) was included in prompts. For data privacy, no proprietary data, code, or non-public results were provided. We avoided uploading full drafts and removed any metadata that could compromise double-blind review.

**Prompts and examples.** Typical prompts included: *"Please copyedit the following paragraph for clarity and brevity without changing technical meaning."* and *"Standardize terminology (task vectors, task arithmetic) and flag any ambiguous phrasing."* The models were instructed not to add facts or alter technical content.

