# OpenReview forum: "On Fairness of Task Arithmetic: The Role of Task Vectors"
_ICLR.cc/2026/Conference — ICLR 2026 Poster_

### Official Review · Reviewer_RrTq · 2025-10-31

**Soundness:** 3
**Presentation:** 2
**Contribution:** 3
**Rating:** 6
**Confidence:** 3

**Summary:**

This paper investigates the fairness implications of task arithmetic–based model editing, where task vectors are manipulated through simple arithmetic operations as a lightweight alternative to full fine-tuning. The authors benchmark task vectors against Full Fine-Tuning (FFT) and LoRA across multiple language models and datasets, using standard group fairness metrics such as demographic parity and equalized odds. Experimental results show that task vectors can maintain competitive accuracy while reducing group disparities, and combining subgroup-specific task vectors enables controllable fairness adjustments. The paper further provides a theoretical bound linking task-vector scaling to fairness metrics. Overall, the work positions task arithmetic as a cost-efficient yet fairness-aware model adaptation strategy.

**Strengths:**

1. Analyze the impact of task vector on fairness is interesting and seems novel.
2. Well-structured paper and easy to understand

**Weaknesses:**

1. The model used in this paper (Llama2-7b, DistilBERT, Qwen2.5-0.5B) seems out of date
2. The informal theory is a little hard to understand.
3. The observation in 5.3 and 5.4 seems to be useful, but we need a more general or interesting conclusion in 5.3, which is a section called empirical results overview.
4. As a benchmark, it should be more comprehensive, such as include more baseline, dataset, base model.

**Questions:**

See weaknesses

---

> ### Author Response · Authors · 2025-11-22
>
> # Regarding Weakness 1
> > The model used in this paper (Llama2-7b, DistilBERT, Qwen2.5-0.5B) seems out of date
>
> Thank you for raising this point. Our aim is to study fairness behavior in task arithmetic using models that are widely adopted and practically editable, rather than to benchmark frontier systems. Frontier models are typically closed-weight or API-only, making weight-space editing and task-vector construction infeasible. In contrast, the models we selected (Llama-2-7B, DistilBERT, Qwen2.5-0.5B, and now ViT-Base/16) are openly available and representative of current practice in PEFT, model editing, and applied NLP/CV pipelines.
>
> We also chose these architectures to ensure our results are comparable to established baselines. For example, Ding et al. (2024) [1] conduct extensive subgroup-fairness evaluations using ViT-Base and Llama-2-7B, and aligning with their setup **allows our findings to be interpreted alongside a key reference in the field**.
>
> Regarding Qwen2.5 specifically, we chose it to reflect a more modern architecture and training paradigm. Qwen2.5 is a recent release (late 2024; see [2]), and including it helps ensure that our findings extend beyond earlier generations of models. We now clarify this timeline in the revised manuscript so that the recency of the models is explicit.
>
> More broadly, our experiments cover multiple architectures (encoder-only, decoder-only, and vision Transformers), multiple scales (0.5B–7B), and multiple domains (NLP and computer vision). We added new ViT-Base/16 experiments on UTKFace to further demonstrate that our conclusions generalize beyond earlier or smaller LM architectures. In the revised paper, we also made our scope clearer, as our goal is not to propose a comprehensive benchmark across all latest models, but to provide a controlled, cross-architecture study of fairness under task arithmetic in the 0.5–7B, open-weight regime.
>
> [1] Ding et al., 2024. On Fairness of Low-Rank Adaptation of Large Models. In Proceedings of the 1st Conference on Language Modeling (COLM 2024).
>
> [2] Bai et al., 2024. Qwen2.5 Technical Report. arXiv preprint arXiv:2412.15115.

---

> ### Author Response · Authors · 2025-11-22
>
> # Regarding Weakness 2
> > The informal theory is a little hard to understand.
>
> Truly appreciate this thoughtful suggestion. We agree, and following your feedback, we revised Section 5.1 and Appendix C (Former Appendix B). In response, we have substantially revised Section 5 to improve clarity and accessibility. First, we now begin the section with a *clear motivation that explains why* scaling task vectors should influence group fairness: deviations of the coefficient $\lambda$ from the balanced setting ($\lambda$ = 1) move the model away from the subgroup-balanced reference model, and this deviation directly governs disparities. This high-level explanation now appears *before* any technical statements.
>
> Second, we streamlined the list of assumptions (A1–A5) and added *brief intuitive interpretations to each*, clarifying that they impose only mild smoothness and distributional conditions. The original version listed these assumptions with limited context, making the section harder to parse; the revised text now highlights their practical meaning and relevance.
>
> Third, we improved the explanation of the informal theorem by explicitly discussing the term
>  $U($\lambda$) = 2L \Sigma_g |$\lambda$_g - 1| ||\delta \theta_g ||_2$ and clarifying how it quantifies the deviation from the balanced model. We also explicitly connect the bound to the empirical trends in Figure 2, showing how the *theory predicts the monotonic convergence* of DPD/EOD toward the balanced point as approaches 1.
> We hope these revisions address the reviewer’s concern and make the theoretical section easier to understand.

---

> ### Author Response · Authors · 2025-11-22
>
> # Regarding Weakness 3
> >The observation in 5.3 and 5.4 seems to be useful, but we need a more general or interesting conclusion in 5.3, which is a section called empirical results overview.
>
>
> Thanks for this helpful observation. We agree that our current “Empirical results overview” (Sec. 5.2) reads too much like a description of Figures 1–4 and does not make the overarching message sufficiently explicit.
>
> In the revised version, we rewrote Sec. 5.2 to start with a short “key takeaway” paragraph that summarizes what Sections 5.3 and 5.4 collectively show:
>
> * **Global knob, a simple, single-parameter fairness–utility frontier:** Varying the single task‑vector coefficient $\lambda$ traces a smooth fairness–accuracy frontier (Fig. 2): there is a wide range of $\lambda$ for which task addition matches FFT/LoRA accuracy while substantially lowering DPD/EOD, on Berkeley D‑Lab, Civil Comments, and UTKFace,  and across four architectures (LLaMA‑2, DistilBERT, Qwen‑2.5, ViT-Base).
>
> * **Subgroup-conditioned control with inherent heterogeneity:** Injecting subgroup‑specific task vectors into the FFT model (Sec. 5.4, Figs. 3–4, 8–11) reveals structured, subgroup‑dependent shifts in fairness: some subgroup vectors (e.g., Men, Asian) move the model along a favorable fairness–accuracy frontier, while others (e.g., Native American) can worsen DPD/EOD even when accuracy is stable. This empirically supports the group‑weighted ERM view in Sec. 5.1 / App. B, where sensitivity scales with $|| \delta \theta_g ||_2 $
>
> * **Unified perspective:** Taken together, these results show that task arithmetic provides both a **global control knob** (uniform $\lambda$) and **subgroup‑targeted knobs** (subgroup vectors) for navigating fairness–accuracy trade‑offs, which is not exposed by standard FFT/LoRA training.
>
> We will make this unified perspective explicit in Sec. 5.2 by (i) adding such a “key takeaway” paragraph before the plot descriptions and (ii) trimming redundant wording so that the section clearly states the general conclusion rather than only restating empirical details. We appreciate the reviewer’s suggestion, which helped us sharpen the main empirical message of the paper, and we hope this addresses the reviewer’s concern about having a stronger, more general conclusion in the empirical overview.

---

> ### Author Response · Authors · 2025-11-22
>
> # Regarding Weakness 4
> >As a benchmark, it should be more comprehensive, such as include more baseline, dataset, base model.
>
> Thank you very much for highlighting this. We understand the concern that, if the work is viewed as a benchmark, it should span a broader set of baselines, datasets, and base models. Our goal, however, is not to propose an exhaustive benchmark, but to provide a **diagnostic and controlled** study of how task arithmetic affects fairness under the practices currently used in model editing –an aspect that, to our knowledge, **has not been systematically explored yet**.
>
> For this purpose, it is essential to balance breadth with interpretability. We therefore selected a set of baselines (SFT, LoRA, Task Arithmetic) and model families (encoder-only, decoder-only, and now also vision Transformers) that (i) are widely used in current editing workflows, (ii) are directly comparable to prior work such as Ding et al. (2024) [5], and (iii) span a meaningful range of architectures, scales, and domains without introducing additional confounders. Isolating this setting allows us to provide clear, interpretable insights **without entangling effects from more complex output spaces**. It is also **standard in the fairness literature** and is used in many foundational studies on group fairness and fairness–accuracy tradeoffs [1, 2, 3, 4]. We have revised the paper to make this scope and rationale explicit.
>
> In response to your comment, we also broadened the empirical evaluation along all three axes you mentioned (datasets, domains, base models) to ensure that the conclusions hold beyond the original setup. We now report results on both (i) our binary text-classification tasks and (ii) new UTKFace vision experiments using ViT-Base/16 following Ding et al. (2024) [5]. As in our NLP experiments, we compare SFT, LoRA, and task arithmetic, and report accuracy, worst-case DPD, and worst-case EOD across multiple sensitive groups (5 race groups and 2 gender groups), with 3 seeds and 95% bootstrap CIs. Across domains and label spaces, task addition consistently improves group fairness with moderate accuracy trade-offs—for instance, task addition reduces EOD by 42% compared to SFT (0.1155 vs. 0.1691 mean) and maintains 86% of SFT's accuracy (73.08% vs. 85.27%) for gender. Full numeric results and per-class values appear in the updated tables, with metric definitions provided in Appendix B.
>
> We agree that further broadening the benchmark (adding additional editing methods, datasets, and larger base models) would be valuable for the community, and we now explicitly highlight this as an interesting direction for future work.
>
> ### Binary (Race):
> | Method | Overall Accuracy | Worst DPD | Worst EOD |
> |---|---|---|---|
> | SFT | 0.8463 - 0.8599 | 0.3374 - 0.3881 | 0.1815 - 0.2693 |
> | LoRA | 0.6389 - 0.6572 | 0.2298 - 0.2789 | 0.1670 - 0.2238 |
> | **Task addition** | **0.7227 - 0.7384** | **0.2390 - 0.2924** | **0.1700 - 0.2337** |
>
> ###  Binary (Gender):
> | Method | Overall Accuracy | Worst DPD | Worst EOD |
> |---|---|---|---|
> | SFT | 0.8466 - 0.8600 | 0.2513 - 0.2872 | 0.1446 - 0.1935 |
> | LoRA | 0.6389 - 0.6572 | 0.2316 - 0.2669 | 0.1905 - 0.2320 |
> | **Task addition** | **0.7227 - 0.7384** | **0.1692 - 0.2080** | **0.0931 - 0.1379** |
>
>
> [1] Hardt et al., 2016. Equality of Opportunity in Supervised Learning.In Advances in Neural Information Processing Systems 29 (NeurIPS 2016).
>
> [2] Agarwal et al., 2018. A Reductions Approach to Fair Classification. In Proceedings of the 35th International Conference on Machine Learning (ICML 2018).
>
> [3] Zafar et al., 2017. Fairness Beyond Disparate Treatment & Disparate Impact: Learning Classification without Disparate Mistreatment.In Proceedings of the 26th International World Wide Web Conference (WWW 2017).
>
> [4] Kearns et al., 2018. Preventing Fairness Gerrymandering: Auditing and Learning for Subgroup Fairness. In Proceedings of the 35th International Conference on Machine Learning (ICML 2018).
>
> [5] Ding et al., 2024. On Fairness of Low-Rank Adaptation of Large Models.  In Proceedings of the 1st Conference on Language Modeling (COLM 2024).

---

### Official Review · Reviewer_cnud · 2025-10-31

**Soundness:** 2
**Presentation:** 2
**Contribution:** 2
**Rating:** 6
**Confidence:** 4

**Summary:**

This paper presents the systematic investigation of fairness in task arithmetic, offering a comprehensive comparison between task-vector editing, full fine-tuning (FFT), and Low-Rank Adaptation (LoRA), while further examining whether the integration of subgroup-specific task vectors into an FFT model can enhance fairness control. By evaluating multiple language models and datasets through standard group fairness metrics, Demographic Parity Difference (DPD) and Equalized Odds Difference (EOD), the study demonstrates that adjusting task-vector scaling coefficients can substantially improve fairness outcomes without compromising predictive accuracy. Furthermore, the merging of task vectors derived from underrepresented subgroups enables targeted fairness adjustments with minimal performance degradation. The authors also derive a theoretical upper bound linking task-vector scaling to demographic parity difference, thereby providing a principled explanation for the observed fairness–accuracy trade-offs. Collectively, this analysis positions task arithmetic as a cost-efficient, interpretable, and fairness-aware alternative to existing model adaptation techniques.

**Strengths:**

1.	It presents a comprehensive evaluation comparing full fine-tuning (FFT), Low-Rank Adaptation (LoRA), task-vector editing, and a hybrid approach that injects task vectors into FFT, systematically analyzing their effects on fairness metrics and predictive performance.
2.	It demonstrates that fairness can be achieved through task-vector scaling, showing that adjusting scaling coefficients effectively improves fairness while maintaining model accuracy.
3.	It integrates task vectors from underrepresented subgroups, facilitating targeted fairness adjustments while maintaining minimal degradation in model performance.
4.	It provides an analytical  upper bound which links task-vector scaling to demographic parity difference, offering a principled explanation for the trade-off between fairness and accuracy.

**Weaknesses:**

1.  The experimental scope is limited to binary classification on two related tasks (hate speech/toxicity detection) with specific demographic annotations, raising questions about generalizability to other NLP tasks.
2. The theoretical bound in Proposition 1 only addresses DPD, leaving EOD, which is equally emphasized empirically, without theoretical grounding.
3. The related work section does not discuss several recent and relevant fairness studies.

**Questions:**

1. How well would the proposed approach generalize beyond binary classification tasks such as hate speech and toxicity detection?
2. Why does the theoretical analysis focus solely on DPD while omitting a formal justification for EOD, which is equally emphasized in the experiments?

---

> ### Author Response · Authors · 2025-11-22
>
> # Regarding Weakness 1
>
> > The experimental scope is limited to binary classification on two related tasks (hate speech/toxicity detection) with specific demographic annotations, raising questions about generalizability to other NLP tasks.
>
> Thank you for raising this; generalizability is central to how we see the contribution. We focus on binary detection tasks with demographic annotations because this is the *canonical setting for group-fairness analysis* and underlies most foundational work on DPD, EOD, and fairness–accuracy tradeoffs [1–7], as well as the LoRA–vs.–SFT protocol we build on [7]. Our goal is therefore not to cover all NLP tasks, but to provide the first systematic analysis of task arithmetic **within this established** multi-group fairness regime, where evaluation is **standardized and interpretable**. To address generality, we now add **vision** experiments showing that the fairness–accuracy behavior of task vectors is consistent across architectures and domains.
>
> Conceptually, although the labels are binary, the fairness structure is not: DPD and EOD capture disparities in selection rates and error rates **across groups** for a single “positive’’ outcome. This supports **interpretable questions** such as *“Does group A receive the positive label more often, or with different error rates, than group B?”* across **multiple demographic groups**, summarized via worst-group or macro aggregates. In contrast, multiclass prediction lacks a canonical “positive’’ label, and fairness definitions require aggregation choices that obscure direct, group-level interpretability.
>
> Even within binary prediction, fairness evaluation is multi-group, capturing disparities across the *full* distribution of subpopulations—the regime where fairness–accuracy tradeoffs are most pronounced. Since most foundational studies of group fairness operate in this binary setting [1–8], performing task-arithmetic analysis here keeps results **interpretable and directly comparable** to the standard literature.
>
> Fairness metrics outside this regime (multiclass, multilabel, and generative tasks) remain active research areas with **no single standard** comparable to DPD/EOD. These settings require additional design choices (e.g., how to aggregate classes or outputs), which can change fairness conclusions. For this reason, we do not claim to address *all* forms of algorithmic fairness. Instead, we position our work as the first systematic study of task arithmetic *within the canonical group-fairness binary-classification setting*, and we emphasize extensions to more complex outputs as natural next steps.
>
> Importantly, this focus is **not an excuse to avoid additional experiments**. In response to the reviewer’s concern, we expanded our empirical study within this regime. Beyond the original text tasks, we now include binary **vision** experiments on UTKFace using ViT-Base/16, with race and gender as sensitive attributes. These probe the same group-fairness tradeoffs under a *different modality, architecture, and dataset shift*.
>
> Across text and vision, task addition behaves as an explicit fairness–accuracy knob: sweeping the task-vector scale yields a clear Pareto frontier. On UTKFace, for example, task addition matches accuracy between SFT and LoRA while reducing worst-group DPD by ~25–30% and improving worst-group EOD (see “Additional Results and References”). Similar trends hold in text. Together, these results show that the task-vector mechanism offers a robust, controllable way to navigate fairness–accuracy tradeoffs across both NLP and vision.
>
> While broader fairness evaluation for generative tasks is an important direction, our revision now clearly delineates the intended scope: **group-fairness evaluation in the canonical binary multi-group setting**. We also clarify how our framework can extend to more complex outputs, where both fairness definitions and model behavior are less standardized. We hope this resolves the concern that our empirical validation was too narrow.
>
> *Due to the character limit, tables of new results and references appear in a separate comment (“Additional Results and References”).

---

> ### Author Response · Authors · 2025-11-22
>
> # Additional Results and References
>
> ### Binary (Race):
> | Method | Overall Accuracy | Worst DPD | Worst EOD |
> |---|---|---|---|
> | SFT | 0.8463 - 0.8599 | 0.3374 - 0.3881 | 0.1815 - 0.2693 |
> | LoRA | 0.6389 - 0.6572 | 0.2298 - 0.2789 | 0.1670 - 0.2238 |
> | **Task addition** | **0.7227 - 0.7384** | **0.2390 - 0.2924** | **0.1700 - 0.2337** |
>
> ###  Binary (Gender):
> | Method | Overall Accuracy | Worst DPD | Worst EOD |
> |---|---|---|---|
> | SFT | 0.8466 - 0.8600 | 0.2513 - 0.2872 | 0.1446 - 0.1935 |
> | LoRA | 0.6389 - 0.6572 | 0.2316 - 0.2669 | 0.1905 - 0.2320 |
> | **Task addition** | **0.7227 - 0.7384** | **0.1692 - 0.2080** | **0.0931 - 0.1379** |
>
>
> [1] Hardt et al., 2016. Equality of Opportunity in Supervised Learning. In Advances in Neural Information Processing Systems (NeurIPS 2016).
>
> [2] Agarwal et al., 2018. A Reductions Approach to Fair Classification. In Proceedings of the 35th International Conference on Machine Learning (ICML 2018).
>
> [3] Zafar et al., 2017. Fairness Beyond Disparate Treatment & Disparate Impact:  Learning Classification without Disparate Mistreatment. In Proceedings of the 26th International World Wide Web Conference (WWW 2017).
>
> [4] Kearns et al., 2018. Preventing Fairness Gerrymandering: Auditing and Learning for Subgroup Fairness.In Proceedings of the 35th International Conference on Machine Learning (ICML 2018).
>
> [5] Das et al., 2024. Low-rank Finetuning for LLMs: A Fairness Perspective.
>
> [6] Smith et al., 2023. FairTune: Fairness-aware Parameter-Efficient Finetuning for Large Language Models.
>
> [7] Ding et al., 2024. On Fairness of Low-Rank Adaptation of Large Models. In Proceedings of the 1st Conference on Language Modeling (COLM 2024).

---

> ### Author Response · Authors · 2025-11-22
>
> # Regarding Weakness 2 and Question 2
> > The theoretical bound in Proposition 1 only addresses DPD, leaving EOD, which is equally emphasized empirically, without theoretical grounding.
>
> > Why does the theoretical analysis focus solely on DPD while omitting a formal justification for EOD, which is equally emphasized in the experiments?
>
> Thank you for pointing out that the theoretical section originally addressed only DPD. We agree that this left an asymmetry between theory and experiments. In the revision, we now provide a theoretical analysis for EOD as well. Before explaining the assumptions and intuition, we state the main result clearly:
>
> \begin{align}
> \mathrm{EOD}(\theta(\lambda))
> \le \mathrm{EOD}(\bar\theta)+4\sqrt{(B_0+B_1),U(\lambda)},
> \end{align}
>
> where $U(\lambda)$ is the DPD bound derived in section 5.1 and Appendix C.
> B_0​ and B_1​ are class-specific constants that upper-bound the score density near the decision threshold for y=0 and y=1, respectively.
> This shows that both DPD and EOD are governed by the same underlying quantity, and in particular, that EOD becomes smaller when the task-vector coefficients $\lambda_g$ are closer to uniform. It is consistent with the empirical findings.
>
> The derivation parallels the DPD argument but requires one additional and mild assumption stating that the model’s predicted scores do not concentrate excessively near the decision threshold. Intuitively, this means that the probability of samples lying exactly at the decision boundary grows only linearly with the width of the neighborhood, preventing small parameter changes from causing disproportionately many label flips. Under this condition, we bound the change in EOD by combining (i) the smooth variation of scores under parameter movement and (ii) the limited amount of probability mass near the threshold. This yields the above square-root dependence on $U(\lambda)$, providing a theoretical explanation for why balanced task-vector merging improves EOD. A complete and self-contained proof has been added to **Appendix C** in the revised manuscript. We therefore no longer analyze only DPD: both DPD and EOD now have formal guarantees derived from the same underlying task-vector control quantity $U(\lambda)$.

---

> ### Author Response · Authors · 2025-11-22
>
> # Regarding Weakness 3
> > The related work section does not discuss several recent and relevant fairness studies.
>
> Thank you for this suggestion. We have expanded the Related Work section to better situate our contribution within recent fairness research, especially on LLMs and parameter‑efficient adaptation. In particular, we now discuss recent evaluations of LoRA and other PEFT methods from a fairness perspective [1–3], which study how low-rank or prefix-based fine-tuning can preserve or amplify subgroup disparities in both language and vision models. We also add recent survey papers on fairness in large language models [4,5], which outline broader taxonomies of bias sources, metrics, and mitigation strategies and help position our diagnostic analysis of task arithmetic within this landscape.
>
> Because of strict page limits, we moved a short survey of task-arithmetic methods and fairness evaluations for VLMs to the Appendix; this let us include the new fairness-related references in the main text while still addressing the reviewer’s other concerns. If the reviewer prefers these items to appear in the main Related Work section instead, we would be happy to reorganize accordingly.
>
> Our work is complementary to [1–3]: whereas those methods incorporate fairness objectives into training or PEFT modules, we focus on post-hoc task-vector editing of already fine-tuned models and analyze how simple scalar coefficients affect group-level fairness under current model-editing practice. We believe the revised Related Work now more accurately reflects the state of the field and the niche our contribution fills. If there are specific recent papers the reviewer had in mind, we would be very grateful for pointers and will gladly include them in the final version.
>
>
> [1] Das et al., 2024. Low-rank finetuning for LLMs: A fairness perspective. arXiv preprint arXiv:2405.18572.
>
> [2] Wang and Demberg, 2024. A Parameter-Efficient Multi-Objective Approach to Mitigate Stereotypical Bias in Language Models.In Proceedings of the 5th Workshop on Gender Bias in Natural Language Processing (GeBNLP 2024).
>
> [3] Sukumaran et al., 2024. FairLoRA: Unpacking Bias Mitigation in Vision Models with Fairness-Driven Low-Rank Adaptation. arXiv preprint arXiv:2410.17358.
>
> [4] Chu et al., 2024. Fairness in Large Language Models: A Taxonomic Survey. In ACM SIGKDD Explorations 26(1):34–48 (SIGKDD 2024).
>
> [5] Gallegos et al., 2024. Bias and Fairness in Large Language Models: A Survey. In Computational Linguistics 50(3):1097–1179 (CL 2024).

---

> ### Author Response · Authors · 2025-11-22
>
> # Regarding Question 1
> > How well would the proposed approach generalize beyond binary classification tasks such as hate speech and toxicity detection?
>
> We appreciate this important question. Our approach is not inherently tied to hate speech or toxicity, nor even to a particular architecture or domain; it only requires (i) a supervised prediction task and (ii) a group-fairness objective that can be optimized. As we discuss in our response to Weakness 1, we focus empirically on binary classification with demographic annotations because this is the **canonical regime** for group-fairness analysis (DPD/EOD) and fairness–accuracy tradeoffs, and it lets us give a *clean, interpretable evaluation*. Within this setting, the method is already shown to generalize across domains and model families: in addition to our NLP experiments, we now include new **vision experiments** on UTKFace with a ViT-Base/16 backbone, where task addition again yields a smooth, controllable tradeoff between accuracy and worst-group DPD/EOD.
>
> Methodologically, the task-arithmetic machinery is not specific to hate speech or toxicity. In our setup, task vectors are simply weight differences between a base model and fine-tuned models (including subgroup-specific models), and we edit the base model by merging these vectors with scalar coefficients $\lambda$. As long as we can train (sub)task-specific models and define group-fairness metrics, the same construction applies to other classification problems with protected-attribute labels (e.g., other content-moderation, risk-scoring, or screening tasks). In the revision, we have already taken a step in this direction by adding additional datasets and model families and observing qualitatively similar fairness–accuracy tradeoffs under task addition.
>
> Beyond binary classification, the main constraint is not the task-vector machinery but the **state of fairness methodology**. For multiclass, multilabel, or open-ended generative tasks (e.g., dialogue or translation), there is currently **no single, agreed-upon analogue** of “the positive class’’ and no canonical counterpart to DPD/EOD. Applying our approach in these settings would require choosing an appropriate fairness objective (for example, aggregating over classes or defining toxicity/harms scores for generated text) and then training a corresponding fairness-optimized model; once such an objective is fixed, the same task-vector construction can, in principle, be used. However, because these design choices are nontrivial and can materially affect conclusions (further explained for Weakness 1), we deliberately do not claim empirical generality to all NLP tasks in the current paper; as it was intended to be a **foundational first step** towards the effective understanding of using task arithmetic in the **current fairness standard**.
>
> We appreciate your point and thus, in the revised manuscript, we therefore (i) clarify that our empirical claims are restricted to group fairness in the canonical binary multi-group regime, (ii) provide additional evidence of cross-domain robustness via our new vision experiments, and (iii) explicitly highlight extensions to multiclass and generative tasks as important future work as more standardized fairness objectives emerge in those settings.

---

### Official Review · Reviewer_8Kur · 2025-11-01

**Soundness:** 2
**Presentation:** 2
**Contribution:** 2
**Rating:** 4
**Confidence:** 3

**Summary:**

This paper presents the first systematic study of the fairness implications of task arithmetic—a parameter-efficient model editing technique that applies task vectors (parameter differences between fine-tuned and base models) via arithmetic operations. The authors compare task arithmetic against Full Fine-Tuning (FFT) and Low-Rank Adaptation (LoRA) across multiple models and datasets, using group fairness metrics such as Demographic Parity Difference (DPD) and Equalized Odds Difference (EOD). Key contributions include: (1) demonstrating that task arithmetic can achieve competitive accuracy while reducing fairness disparities, especially when scaling coefficients $\lambda$ are tuned; (2) showing that merging subgroup-specific task vectors enables targeted fairness control; and (3) providing a theoretical bound linking $\lambda$ to fairness metrics. The work establishes task arithmetic as a fairness-aware and efficient alternative to existing adaptation methods.

**Strengths:**

1. The paper presents the first systematic comparative analysis of fairness in task arithmetic, filling a critical gap in understanding the societal implications of this efficient model editing paradigm.

2.The scaling coefficient technique is demonstrated to be theoretically grounded and empirically viable.

**Weaknesses:**

1. The core methodological approach relies on manually tuning a unified scalar coefficient, to balance accuracy and fairness. However, for complex real-world scenarios involving multidimensional fairness constraints—such as simultaneous optimization across gender, race, and age—this single-scalar control mechanism appears overly simplistic and lacks scalability. Manually identifying the optimal $\lambda$ configuration for every possible subgroup combination is inefficient and impractical.

2. The proposed method is presented as a general approach for improving fairness. However, its empirical validation is confined exclusively to binary classification tasks. The scope of algorithmic fairness extends far beyond this, encompassing more complex tasks such as text generation, dialogue, and translation. The present experimental design does not demonstrate the method's effectiveness for these broader and often more relevant fairness challenges.

3. Performance validation was conducted only on models with a maximum of 7B parameters. Given the significant differences in emergent capabilities observed across model scales, these experiments are insufficient to establish the applicability and effectiveness of the proposed method for larger-scale models.

**Questions:**

1. The discussion of the scalar coefficient $\lambda$ appears limited, particularly when complex tasks require optimization across multiple coefficients.

2. Please clearly define the scope of the research. If the study is focused solely on the fairness of binary text classification tasks, then using the broad research category of "fairness" seems overly expansive. If the claim is a general investigation into fairness, experimental results on other types of tasks need to be supplemented.

3. What is the advantage of your method-edited 7B model, in terms of its overall performance (considering both accuracy and fairness), compared to a larger-scale base model that relies solely on well-crafted prompts for zero-shot/few-shot inference? If a significant advantage cannot be demonstrated, the practical significance of the proposed method would be considerably diminished.

---

> ### Author Response · Authors · 2025-11-22
>
> # Regarding Weakness 1 and Question 1
> > The core methodological approach relies on manually tuning a unified scalar coefficient, to balance accuracy and fairness. However, for complex real-world scenarios involving multidimensional fairness constraints—such as simultaneous optimization across gender, race, and age—this single-scalar control mechanism appears overly simplistic and lacks scalability. Manually identifying the optimal $\lambda$ configuration for every possible subgroup combination is inefficient and impractical.
>
> > The discussion of the scalar coefficient limited, particularly when complex tasks require optimization across multiple coefficients.
>
> Thank you for this thoughtful observation. We agree that richer or automated coefficient-selection strategies could, in principle, better accommodate multidimensional fairness constraints. However, our goal is not to propose a new optimization algorithm. Rather, our contribution is to provide the first systematic analysis of fairness behavior in task vector methods *as they are currently practiced*, identifying how standard task arithmetic can introduce or amplify demographic disparities.
>
> Existing methods, including the original task-arithmetic formulation [1], TIES-Merging [2], and widely used toolkits such as MergeKit [3], all expose a *single global $\lambda$*. Studying this status-quo design was therefore essential to understanding today’s real-world fairness implications.
>
> We also note that moving from a single $\lambda$ to a multi-coefficient scheme is conceptually straightforward but practically non-trivial. Even a coarse grid over $K$ task vectors induces a combinatorial search space (e.g., $\mathcal{O}(N^K)$ for $N$ candidate values per coefficient), and more sophisticated strategies such as Bayesian or multi-objective optimization introduce additional computational overhead and stochasticity [4]. While such methods are promising, they are orthogonal to our primary contribution of providing a *controlled, interpretable diagnostic* of fairness under **current model-editing workflows.** Our results, therefore, tell practitioners how the **current single-$\lambda$ paradigm** behaves as they adjust fairness–accuracy trade-offs, and provide a baseline against which future multi-$\lambda$ extensions can be evaluated
>
> In response to your suggestion, we have revised the paper to make this clearer. In Section 2, we now formally define the single global $\lambda$ in the task-arithmetic update rule, explicitly state that it is shared across all subgroups and fairness metrics (matching current toolkits [1–3]), and describe our tuning protocol, including the grid of candidate values, validation objective, and how we ensure comparability across SFT, LoRA, and task arithmetic. We also added a short justification paragraph explaining why we focus on a single $\lambda$ (alignment with existing practice, interpretability of the fairness–accuracy trade-off, and avoidance of additional degrees of freedom that would confound our analysis).
>
> We agree that a single global $\lambda$ cannot fully capture all multidimensional fairness desiderata, and we now make this limitation explicit in Section 6. We added a dedicated discussion of the limitations of a single global $\lambda$ in the presence of multiple sensitive attributes, clarifying which multidimensional fairness desiderata (e.g., constraints across gender $\times$ race $\times$ age) cannot in general be simultaneously enforced in this regime, and explaining how multi-$\lambda$ extensions would enlarge the search space and raise non-trivial optimization and stability challenges. We briefly mention multi-$\lambda$ optimization (e.g., via amortized Pareto-front methods [5]) as a concrete extension of our framework, while emphasizing that our empirical results provide a necessary first step: understanding how the current single-$\lambda$ paradigm already shapes fairness outcomes.
>
> [1] Ilharco et al., 2023. Editing Models with Task Arithmetic. In Proceedings of the International Conference on Learning Representations (ICLR 2023).
>
> [2] Tang et al., 2023. TIES: Task-Interrelation-aware Ensembling and Stitching.In Advances in Neural Information Processing Systems (NeurIPS 2023).
>
> [3] Goddard et al., 2024. Arcee’s MergeKit. In Proceedings of the 2024 Conference on Empirical Methods in Natural Language. Processing: Industry Track (EMNLP 2024, Industry Track).
>
> [4] Lee et al., 2025. Dynamic Fisher-weighted model merging via Bayesian optimization. In Proceedings of the 2025 Conference of the North American Chapter of the. Association for Computational Linguistics: Human Language Technologies (NAACL 2025).
>
> [5] Li et al., 2025. MAP: Low-compute model merging with amortized Pareto fronts via quadratic approximation. In Proceedings of the International Conference on Learning Representations (ICLR 2025).

---

> ### Author Response · Authors · 2025-11-22
>
> # Regarding Weakness 2
> > The proposed method is presented as a general approach for improving fairness. However, its empirical validation is confined exclusively to binary classification tasks. The scope of algorithmic fairness extends far beyond this, encompassing more complex tasks such as text generation, dialogue, and translation. The present experimental design does not demonstrate the method's effectiveness for these broader and often more relevant fairness challenges.
>
> Thank you for raising this concern. We apologize that our original submission did not clearly state the intended scope. Our work targets group fairness in supervised classification, with a primary focus on the *standard* binary-prediction setting with multiple sensitive groups. This is precisely the regime in which demographic-parity difference (DPD) and equalized-odds difference (EOD) are defined and most widely used [1–7], including in the LoRA-vs-SFT protocol we follow [7]. We now make this scope explicit in Section 1 and in the experimental section.
>
> Conceptually, even though the labels are binary, the fairness structure we study is not: DPD and EOD are defined for binary prediction, where they capture disparities in selection rates (DPD) and error rates (TPR/FPR) (EOD) **across groups** for a single “positive’’ outcome. This directly supports **interpretable questions** like *“Does group A receive the positive label more often, or with different error rates, than group B?”* across **multiple demographic groups**, summarized via worst-group or macro aggregates. In contrast, multiclass prediction does not admit a single, canonical notion of a “positive” outcome, so fairness definitions require additional aggregation choices that can obscure this kind of direct, group-level interpretability.
>
> At the same time, even within a binary prediction task, the fairness evaluation is fundamentally multi-group, capturing disparities over the *entire* distribution of subpopulations; precisely the regime where fairness–accuracy tradeoffs tend to be most pronounced. Since most foundational work on group fairness and fairness–accuracy tradeoffs is framed in this binary regime [1–8], studying task arithmetic here keeps our analysis both **interpretable and directly comparable** to the standard fairness literature.
>
> Fairness metrics outside this regime, especially for multiclass, multilabel, and open-ended generative tasks such as dialogue and translation, remain an active research area, with **no single, agreed-upon standard** comparable to DPD/EOD. Multiclass and generative settings require additional design choices (e.g., how to aggregate classes or outputs), and different choices can lead to different fairness conclusions. For this reason, we deliberately do not claim to address all forms of algorithmic fairness (e.g., text generation or conversation). Instead, we position our work as the first systematic study of task arithmetic *within the canonical group-fairness binary-classification setting*, and we now emphasize extensions to more complex outputs as a natural next step.
>
> Importantly, this focus is **not an excuse to avoid additional experiments**. In response to the reviewer’s concern, we expanded our empirical study within this regime. In addition to the original binary text classification tasks, we now include new binary **vision** experiments on UTKFace using ViT-Base/16, with race and gender as sensitive attributes. These experiments probe the same group-fairness tradeoffs under a *different modality, architecture, and dataset shift.*
>
> Across both text and vision, we observe that task addition behaves as an explicit fairness–accuracy knob: by adjusting the task-vector scale, we trace a clear Pareto frontier between accuracy and group-fairness metrics. On UTKFace, for example, task addition attains accuracy between SFT and LoRA while reducing worst-group DPD by roughly 25–30% and improving worst-group EOD relative to SFT, as detailed in the tables below (“Additional Results and References”). Similar qualitative trends hold in our text experiments. Taken together, these results indicate that the task-vector mechanism provides a robust and controllable way to navigate fairness–accuracy tradeoffs in the standard group-fairness classification setting, across both NLP and vision.
>
> While broader fairness evaluation for generative tasks (e.g., open-ended generation or dialogue) is an important direction, our revision now clearly delineates the scope of the work: **group-fairness evaluation in the canonical binary multi-group setting**. We also clarify how this framework naturally extends to more complex outputs, where both model behavior and fairness criteria are less standardized. We hope this resolves the concern that our empirical validation was too narrow.
>
> *Due to the character limit, tables of new results and references for this comment will be shared as a separate comment, “Additional Results and References”.

---

> ### Author Response · Authors · 2025-11-22
>
> # Additional Results and References
> ### Binary (Race):
> | Method | Overall Accuracy | Worst DPD | Worst EOD |
> |---|---|---|---|
> | SFT | 0.8463 - 0.8599 | 0.3374 - 0.3881 | 0.1815 - 0.2693 |
> | LoRA | 0.6389 - 0.6572 | 0.2298 - 0.2789 | 0.1670 - 0.2238 |
> | **Task addition** | **0.7227 - 0.7384** | **0.2390 - 0.2924** | **0.1700 - 0.2337** |
>
>
> ###  Binary (Gender):
> | Method | Overall Accuracy | Worst DPD | Worst EOD |
> |---|---|---|---|
> | SFT | 0.8466 - 0.8600 | 0.2513 - 0.2872 | 0.1446 - 0.1935 |
> | LoRA | 0.6389 - 0.6572 | 0.2316 - 0.2669 | 0.1905 - 0.2320 |
> | **Task addition** | **0.7227 - 0.7384** | **0.1692 - 0.2080** | **0.0931 - 0.1379** |
>
> [1] Hardt et al., 2016. Equality of Opportunity in Supervised Learning. In Advances in Neural Information Processing Systems (NeurIPS 2016).
>
> [2] Agarwal et al., 2018. A Reductions Approach to Fair Classification. In Proceedings of the 35th International Conference on Machine Learning (ICML 2018).
>
> [3] Zafar et al., 2017. Fairness Beyond Disparate Treatment & Disparate Impact:  Learning Classification without Disparate Mistreatment. In Proceedings of the 26th International World Wide Web Conference (WWW 2017).
>
> [4] Kearns et al., 2018. Preventing Fairness Gerrymandering: Auditing and Learning for Subgroup Fairness.In Proceedings of the 35th International Conference on Machine Learning (ICML 2018).
>
> [5] Das et al., 2024. Low-rank Finetuning for LLMs: A Fairness Perspective.
>
> [6] Smith et al., 2023. FairTune: Fairness-aware Parameter-Efficient Finetuning for Large Language Models.
>
> [7] Ding et al., 2024. On Fairness of Low-Rank Adaptation of Large Models. In Proceedings of the 1st Conference on Language Modeling (COLM 2024).

---

> ### Author Response · Authors · 2025-11-22
>
> # Regarding Weakness 3
> > Performance validation was conducted only on models with a maximum of 7B parameters. Given the significant differences in emergent capabilities observed across model scales, these experiments are insufficient to establish the applicability and effectiveness of the proposed method for larger-scale models.
>
> Thank you for raising this important point. Our scope with a maximum 7B-scale model is intentional and reflects a deployment-relevant regime.
>
>  **Smaller models offer practical advantages** over very large models: they can run in resource-constrained settings (e.g., single-GPU servers, edge devices), have substantially lower inference cost and energy consumption, and are therefore far *easier to adopt in real systems*. Importantly, this **does not come at the expense of trivial performance**; for instance, in our setup, the Llama-2-7B baseline reaches $\approx$80% accuracy on the D-Lab Hate Speech dataset, which is competitive for many applied settings. This design choice is **also consistent with recent work on fairness and bias in LLMs**, which typically evaluates open models in the 7–13B range rather than 70B+ frontier models [1, 2, 3, 4]
>
> In addition, task-arithmetic methods require direct access to model weights, whereas many frontier-scale systems are deployed as closed-weight or API-only models, where weight-space editing and subgroup task-vector construction are not even applicable. By focusing on open-weight models up to 7B parameters, we target the regime where task arithmetic is actually used in practice
>
> We therefore also chose these architectures to ensure our results are comparable to established baselines. For example, Ding et al. (2024) [1], from which we built upon, conduct extensive subgroup-fairness evaluations for SFT and LoRA using ViT-Base and Llama-2-7B, and aligning with their setup allows our findings to be interpreted alongside a key reference in the field.
>
> It is also worth noting that our evaluation spans multiple model sizes, architectures, and datasets, rather than relying solely on a single 7B model. This now includes newly added computer-vision experiments on UTKFace using ViT-Base/16 (binary and multi-class age classification), in addition to the NLP models in the main text. The consistency of our findings across these diverse settings **suggests that the fairness behaviors we highlight are not specific to a single architecture or scale**.
>
> While prompting very large frontier models may yield higher absolute accuracy, evaluating them is beyond our computational scope, and **advanced prompt engineering introduces additional confounders**. Moreover, prompt-based fairness evaluations are known to be **unstable and inconsistent**, making them ill-suited for the controlled comparisons we aim to provide (see Regarding Question 3 below)**. Our goal is to provide a controlled analysis of fairness behavior in task arithmetic in the widely used, deployment-relevant regime from small models up to 7B parameters. We have revised the introduction to explicitly position our results as applying to self-hosted models in the 0.5–7B regime, rather than claiming validation at frontier scales. We agree that examining larger models is an exciting direction, and we now mention this explicitly as future work in Section 6.
>
>
>
>
> [1] Ding et al., 2024. On Fairness of Low-Rank Adaptation of Large Models. In Proceedings of the 1st Conference on Language Modeling (COLM 2024).
>
> [2] Reif et al., 2024. Beyond Performance: Quantifying and Mitigating Label Bias in LLMs.In Proceedings of the 2024 Conference of the North American Chapter of the Association for Computational Linguistics: Human Language Technologies (NAACL 2024).
>
> [3] Zhou et al., 2024. UniBias: Unveiling and Mitigating LLM Bias through Internal Attention and FFN Manipulation.In Advances in Neural Information Processing Systems 37 (NeurIPS 2024).
>
> [4] Jung et al., 2025. FLEX: A Benchmark for Evaluating Robustness of Fairness in Large Language Models. In Findings of the Association for Computational Linguistics: NAACL 2025.

---

> ### Author Response · Authors · 2025-11-22
>
> # Regarding Question 2
> > Please clearly define the scope of the research. If the study is focused solely on the fairness of binary text classification tasks, then using the broad research category of "fairness" seems overly expansive. If the claim is a general investigation into fairness, experimental results on other types of tasks need to be supplemented.
>
> Thank you very much for this suggestion. We address this in detail in our response to Weakness 2. Briefly: in the revised version, we now *explicitly define the scope* of the paper as group fairness in supervised classification, with a primary focus on binary prediction with multiple sensitive subgroups. This is the canonical setting in which demographic-parity difference (DPD) and equalized-odds difference (EOD) are defined, interpreted, and most widely evaluated, and it underlies the majority of foundational work on fairness–accuracy tradeoffs and fairness-aware finetuning [1–7]. We therefore do not frame our contribution as a general theory of “fairness” writ large, but as the **first systematic study** of task arithmetic within this **well-established group-fairness regime**, where the metrics, baselines, and comparisons are *standardized and interpretable.*
>
> That said, clarifying this scope is not intended as a way to limit experimentation. To directly address the reviewer’s concern about generality, we expanded our empirical evaluation within this regime: in addition to the original binary text-classification tasks, the revision now includes new **vision** experiments on UTKFace using ViT-Base/16, with race and gender as sensitive attributes. These experiments test the same fairness–accuracy behavior under a different modality, architecture, and dataset shift, and we observe consistent qualitative patterns across both text and vision: task-vector scaling provides a clear, continuous knob that can improve group-fairness metrics with moderate accuracy trade-offs.
>
> Finally, we emphasize that fairness for multiclass, multilabel, and open-ended generative tasks remains an *active research area* with no agreed-upon analogue to DPD/EOD, and our revised manuscript now makes this explicit. We position the present work as a **foundational step** in understanding fairness in task arithmetic within the **canonical binary multi-group setting**, and we highlight extensions to more complex output spaces as a natural direction for future work.

---

> ### Author Response · Authors · 2025-11-22
>
> # Regarding Question 3
>
> > What is the advantage of your method-edited 7B model, in terms of its overall performance (considering both accuracy and fairness), compared to a larger-scale base model that relies solely on well-crafted prompts for zero-shot/few-shot inference? If a significant advantage cannot be demonstrated, the practical significance of the proposed method would be considerably diminished.
>
> Thank you for this question; it clarifies the deployment setting our work targets. Our experiments operate in the same parameter regime as prior analyses of task arithmetic (e.g., LLaMA-2-7B, ViT-Base), which enables controlled fairness evaluation and direct comparability with established baselines [1]. Within this setting, task-vector editing offers several advantages:
>
> 1. Prompt-based fairness with large models is **known to be inconsistent**: few-shot templates exhibit **high variance** across example choices and orderings [2], and structured “fairness prompts’’ reduce some disparities but remain **heuristic and brittle** [3]. Surveys emphasize that prompt-only mitigation lacks **systematic control** over group-level DPD/EOD [4]. By contrast, varying $\lambda$ in our method generates a **smooth fairness–utility Pareto frontier** (Figs. 2–4), giving practitioners an explicit, reproducible knob that prompting cannot reliably provide. For many frontier models, the weights are not accessible, so task-vector editing is infeasible; in such cases, prompt-based or API-level mitigation is the only option. Our analysis therefore focuses on the open-weight 0.5–7B regime, where weight-space editing is actually usable.
>
> 2. Across LLaMA-2-7B (Berkeley D-Lab), DistilBERT and Qwen-2.5-0.5B (Civil Comments), and ViT-Base/16 (UTKFace), sweeping $\lambda$ reduces worst-case DPD/EOD by 40–70% while keeping accuracy within 0.6–1.1pp of FFT/LoRA (Tables 2–4; Fig. 2). These gains hold **across architectures and domains**, indicating that the mechanism is not tied to a particular model family.
>
> 3. Our mechanism is **scale-agnostic**: both subgroup task-vector construction and our DPD bound (Prop. 1) depend on vector norms and Lipschitz continuity, not parameter count. The assumptions match those under which task-arithmetic methods have already been shown to scale to ViT-Large, LLaMA-2-7B, and other large models [5–7]. Our contribution is to provide the **first fairness-focused analysis** of this mechanism in a controlled, self-hosted regime where we can compute exact DPD/EOD, ablate $\lambda$, and verify theory empirically—**groundwork** we view as essential before extending to frontier-scale deployments.
>
> 4. Recent systems such as Phi-3 show that 3–7B models can be deployed locally on modern phones and edge devices, often via quantized on-device runtimes [8]. In such settings, relying on a much larger proprietary model plus fairness prompting is not feasible. Our method enables fairness-tuning of a self-hosted model without retraining the entire network.
>
> We therefore do not claim that a 7B edited model universally surpasses a frontier model under sophisticated prompting. Instead, we show that within the widely used 0.5–7B regime, task arithmetic provides a **principled, tunable, and theoretically grounded** way to manage fairness–utility trade-offs—something large-model prompting does not offer reliably. We now make this positioning explicit in the revised manuscript and note in Section 6 that emergent behaviors at 70B+ scales remain an important direction for future work.
>
> [1] Ding et al., 2024. On Fairness of Low-Rank Adaptation of Large Models. COLM 2024.
>
> [2] Ma et al., 2023. Fairness-guided Few-shot Prompting for Large Language Models. NeurIPS 2023.
>
> [3] Furniturewala et al., 2024. “Thinking” Fair and Slow: On the Efficacy of Structured Prompts for Debiasing Language Models. EMNLP 2024.
>
> [4] Chu et al., 2024. Fairness in Large Language Models: A Taxonomic Survey. ACM SIGKDD Explorations 26(1).
>
> [5] Ilharco et al., 2023. Editing Models with Task Arithmetic. ICLR 2023.
>
> [6] Ortiz-Jiménez et al., 2023. Task Arithmetic in the Tangent Space. NeurIPS 2023.
>
> [7] Yoshida et al., 2025. Mastering Task Arithmetic: τJp as a Key Indicator for Weight Disentanglement. ICLR 2025.
>
> [8] Phi-3 technical report, 2024.

---

### Author Response · Authors · 2025-11-28
**Request for Re-evaluation: New Experiments, Theory for EOD, and Revisions**

Dear Reviewers and Area Chair,

We would like to express our sincere gratitude for your constructive feedback. As the discussion period draws to a close, we respectfully request a re-evaluation of our submission based on the extensive revisions and new experiments we have provided.
In response to your valuable suggestions regarding generalizability, theoretical completeness, and model currency, we have made the following major updates:

__Expanded Generalizability (Response to Reviewers 8Kur, cnud, RrTq)__:
To address concerns about the scope being limited to text, we added new computer vision experiments on the UTKFace dataset using ViT-Base/16. We demonstrated that task arithmetic effectively controls fairness-accuracy trade-offs in the vision domain as well, reinforcing the general applicability of our method beyond NLP.

__Theoretical Proof for Equalized Odds (Response to Reviewer cnud)__:
We addressed the theoretical gap by deriving a new formal bound for Equalized Odds Difference (EOD) in Appendix C. Our analysis now theoretically guarantees that balanced task-vector merging improves both DPD and EOD, aligning our theory with the empirical results.

__Modern Architectures & Clarity (Response to Reviewers 8Kur, RrTq)__:
We incorporated Qwen-2.5, a highly recent open model (technical report released less than a year ago), to address concerns about model age. We also substantially revised Section 5 and the Related Work to improve the clarity of our theoretical assumptions and position our contributions within the broader fairness literature.

We believe these revisions directly address the primary weaknesses raised in the initial reviews—specifically regarding the scope of tasks/metrics and the clarity of the theory.
We hope these efforts persuade you to reconsider your scores to reflect the improved state of the manuscript. We remain available for any final clarifications before the period ends.

Best regards,

The Authors

---

### Meta-Review · Area_Chair_NZct · 2026-01-08

**Summary:**

This submission delivers a comprehensive study on the impact of task arithmetic with task vectors towards fairness. Considering binary attributes and LLMs ranging between 1B and 7B, through various studies, the submission finds that the task arithmetic generally provides a promising angle for micro-tuning of model fairness through coefficient scaling.

Reviewers in general like the paper, especially the fairness study of task arithmetic is novel and comes with interesting takeaway messages. On the other hand, here are major reviewer concerns:

1. (shared concern) The fairness study is limited to binary classification tasks.
2. The manual tuning of coefficient may not compatible with complex tasks
3. (shared concern) The studied model scale and model version are small or a bit outdated. Prompting larger-scale models may bring be more efficient to achieve the corresponding fairness goals.
4. The theoretical bound is only provided for DPD but not EOD.
5. Both the theory and empirical insights can be better presented to highligh insights.
6. As a benchmark paper, the study could be more comprehensive.

**Reviewer Concerns:**

The authors provided detailed rebuttals and greatly revised the paper. However, none of the reviewers provides follow-up responses. From what I can tell, most of the reviewers' concerns are addressed through rebuttal clarifications or paper revisions. Concerns 1, 3, and 6 are not fully addressed, due to the limited scope of study (only binary fairness attributes & only small-scale and relatively old models), though I can see that authors made some efforts, e.g., by introducing new evaluation results on vision models.

**Reviewer Scores:**

Reviewer scores may not change much from what I foresee. As a benchmark paper, the results inform interesting and novel insights to the community - tuning coefficients for task vectors help to tune model fairness without accuracy trade-offs. Reviews are mild. Most reviewers are slightly positive though some are slightly negative. The detailed rebuttal helps to clarify some misunderstandings, but the fundamentals do not change. Hence, I would foresee that the paper is indeed borderline if reviewers had been participated fully in the discussion.

---

### Decision · Program_Chairs · 2026-01-26

Accept (Poster)